

# Merging modelled and reported flood impacts in Europe in a combined flood event catalogue, 1950-2020

Dominik Paprotny[1], Belinda Rhein[1,2], Michalis I. Vousdoukas[3], Paweł Terefenko[4], Francesco Dottori[5], Simon Treu[1], Jakub Śledziowski[4], Luc Feyen[6], and Heidi Kreibich[7]

[1] Potsdam Institute for Climate Impact Research (PIK), Member of the Leibniz Association, P.O. Box 60 12 03, 14412 Potsdam, Germany, [2] Humboldt-Universität zu Berlin, Berlin, Germany, [3] University of the Aegean, Department of Marine Sciences, Mytilene, Greece, [4] Institute of Marine and Environmental Sciences, University of Szczecin, Adama Mickiewicza 16, 70-383 Szczecin, Poland, [5] CIMA Research Foundation, Savona, Italy, [6] European Commission, Joint Research Centre (JRC), Ispra, Italy, [7] GFZ German Research Centre for Geosciences, Section Hydrology, Potsdam, Germany
*Correspondence to*: Dominik Paprotny (dominik.paprotny@pik-potsdam.de)

**Abstract.** Long-term trends in flood losses are regulated by multiple factors including climate variation, demographic dynamics, economic growth, land-use transitions, reservoir construction and flood risk reduction measures. Attribution of those drivers through the use of counterfactual scenarios of hazard, exposure or vulnerability first requires a good representation of historical events, including their location, intensity and the factual circumstances in which they occurred. Here, we develop a chain of models that is capable of recreating riverine, coastal and compound floods in Europe between 1950 and 2020 that had a potential to cause significant socioeconomic impacts. This factual catalogue of almost 15,000 such events was scrutinised with historical records of flood impacts. We found that at least 10% of them had led to significant socioeconomic impacts (including fatalities) according to available sources. The model chain was able to capture events responsible for 96% of known impacts contained in the HANZE flood impact database in terms of persons affected and economic losses, and for 81% of fatalities. The dataset enables studying drivers of vulnerability and flood adaptation due to a large sample of events with historical impact data. The model chain can further be used to generate counterfactual events, especially related to climate change and human influence on catchments.

## 1 Introduction

Flood risk is constantly evolving and influenced by a wide array of drivers, related to atmospheric, land surface and socio-economic processes (Merz et al., 2021). Recent decades have been identified as a particularly flood-rich period along European rivers (Blöschl et al., 2020) and increasing sea levels are expected to exacerbate coastal flood risk (Vousdoukas et al., 2017, 2023, Nicholls et al., 2021). At the same time exposure is growing rapidly (Paprotny et al., 2018b, Rentschler et al., 2023) and mitigation actions are implemented in reaction to floods (Kreibich et al., 2022). Disentangling the different risk drivers requires considerable modelling effort to reconstruct the factual circumstances surrounding the occurrence of floods and modelling





them again under alternative (counterfactual) conditions (Scussolini et al., 2023). Such analyses enable impact attribution, i.e.
linking changes in impacts with their likely causes. It can then provide information on long-term development of risk, which
in turn has implications on cost-benefit analyses or risk management planning (Kreibich et al., 2019).
The recent Sixth Assessment Report of the Intergovernmental Panel on Climate Change, in the chapter on Europe (Bednar-
Friedl et al., 2022), indicated low confidence in trends in riverine and coastal flood impacts in the past half-century, even if
some increase was detected for parts of the continent. The report contained very limited information on attribution, but this
gap is being slowly filled by new studies. For example, Sauer et al. (2021) quantified hazard, exposure and vulnerability
changes for flood events globally, finding that for Europe the increase in flood losses was driven almost entirely by exposure,
with some small decline in hazard and vulnerability. Though the timeframe of the study was short (1980–2010), it highlighted
the role of exposure similarly to Paprotny et al. (2018b), who presented exposure-adjusted losses for 1870–2016 (with
consideration for gaps in flood impact reporting), finding no upward trend in economic losses and a strong decline in fatalities.
Long-run global data on climatic and socioeconomic drivers under factual and counterfactual scenarios are available from the
Inter-Sectoral Model Intercomparison Project, or ISIMIP 3a (Frieler et al., 2024), but they mostly have coarse resolution that
is not easily applicable to Europe and have not yet been used for flood impact attribution. Impact attribution of European
floods was also carried out with a case study-based, semi-quantitative approach of comparing "paired events", i.e. floods that
have occurred in the same area some years apart (Kreibich et al., 2023). This approach has an advantage mainly in the context
of drawing practical conclusions for flood adaptation (Kreibich et al., 2019). Studies that derived projections of future flood
risk in Europe have indicated that all three components of risk play an important role in determining changes in the impact
magnitude (Rojas et al., 2013, Vousdoukas et al., 2018, Steinhausen et al., 2022, Schoppa et al., 2024).
Particular effort is needed in reconstructing the intensity and spatial footprint of flood events. For instance, the loss-
normalisation study of Paprotny et al. (2018b) used 100-year riverine and coastal flood hazard maps as proxies for impact
zones within subnational regions indicated as affected in the HANZE database (Paprotny et al., 2018a). This approach did not
include the effect of climate change, human influence on catchments or simply the variation in return period of different events.
There have been attempts to reconstruct past river floods for North America (Wing et al., 2021) or storm surge footprints
globally (Enríquez et al., 2020), but none specifically for Europe. Satellite-derived flood footprints can also be linked to impact
records, as in Mester et al. (2023), but such datasets cover only a short timeframe and do not resolve the problem of generating
a counterfactual hazard scenario.
In this study we develop a modelling chain to generate a factual flood catalogue for 42 European countries covering the period
1950–2020, which could be further used to run counterfactual scenarios. We only cover the factual scenarios and focus on
deriving the best possible reconstruction of past riverine, coastal and compound floods. The main metric of success of the
modelling chain is its ability to correctly derive the time, location and intensity of 2037 actual floods contained in the HANZE
flood impact database (Paprotny et al., 2023). We further aim at deriving not only the floods that caused significant
socioeconomic impacts, but also those that did not happen despite their hydrological extremity due to existing flood protection,
as this could later be used to quantify the level of European flood protection.



Thanks to the availability of new high-resolution estimates of past population and economic exposure (Paprotny and Mengel,
2023), we narrow down our catalogue of floods only to those with significant socioeconomic impact potential, rather than
those which were extreme only from a hydrological perspective. This enables comparison with historical records of flood
impacts and classifying the modelled events in accordance to their real-life consequences (or lack thereof). Finally, the focus
is on coastal, compound and slow-onset riverine flooding. Flash flood events occurring in small catchments (i.e. with an
upstream area below 100 km²) are not considered in our analysis due to the insufficient resolution of the riverine flood models
available for Europe. Furthermore, we explicitly omit urban floods resulting from insufficient storm drainage rather than from
channel overflow.
The paper provides a short method overview in section 2.1, which is followed by details on the coastal (2.2) and riverine (2.3)
components of the modelling chain, which are brought together for a final flood catalogue compared with historical records
(2.4). Validation of the hydrological hazard follows in the next sections (2.5, 3.1), with an overview of risk indicators derived
from the catalogue (3.2) and finally comparison between modelled and observed flood impacts (3.3). The discussion analyses
the limitations and uncertainties of both the modelled (4.1) and observational data (4.2),, before drawing conclusions and
highlighting possible applications of the flood catalogue (section 5).
**2 Methods**
**2.1 Overview**
Simulating riverine and coastal floods requires different modelling approaches. First we derive extreme river discharges and
coastal water levels, then we apply a common approach  to produce flood intensity maps, compute damages, and aggregate
the results spatiotemporally. Compound floods are generated by combining the results of the two strands of modelling work,
therefore we run the coastal model first, and compound floods are considered as  part of the riverine component, drawing on
the previous coastal results. The methodology is briefly summarised in Table 1.



**Table 1. Summary of the methodology, sections with the corresponding descriptions are given in square brackets.**

| Step | Coastal floods | Riverine floods |
|---|---|---|
| Input climate data | ERA5, interpolated to 0.11° rotated grid **[2.2.1]** | ERA5-Land, statistically downscaled and bias adjusted to 1 arc min **[2.3.1]** |
| Hydrodynamic simulations | Delft3D storm surge height simulation with external data for total water level (hourly) **[2.2.2]** | LISFLOOD river discharge (Q) simulation (6-hourly) **[2.3.2]** |
| Deriving extreme events | TWL > 99.6th percentile per coastal segment **[2.2.3]** | Q > 98th percentile per river grid cell **[2.3.3]** |
| | Merging in time (+/- 2 days) and space (NUTS3 regions), including derivation of compound events **[2.2.3, 2.3.3]** | |
| Estimating flood footprint | If TWL/Q > 2-year return period, water depth interpolated from a set of flood hazard maps (100 m resolution) **[2.2.4, 2.3.4]** | |
| Aggregating flood events | All significantly affected NUTS3 regions aggregated per country using +/- one day window **[2.4.1]** | |
| Potential loss modelling | Persons affected, fatalities and economic loss computed from 100 m exposure grid and damage functions **[2.4.1]** | |
| Filtering and analysis of flood events | Removing events with low potential impacts Intersecting remaining events with historical impact databases **[2.4.2]** Comparing modelled event catalogue with other historical records **[2.4.3]** | |
| Validation | River discharges, storm surge heights and flood footprints **[2.5, 3.1]** | |
| Further analysis | See sections **3.2 and 3.3** | |


In Table 1, the aggregation of extreme discharge or water levels spatially by using NUTS3 regions is mentioned. This refers
to the European Union's (EU) Nomenclature of Territorial Units for Statistics (NUTS). This classification has 4 levels (0, 1,
2, 3), in which 0 is the national level and 3 is the finest sub-regional division. NUTS3 regions are usually administrative
divisions, though at times statistical (analytical) regions are used instead, by amalgamating smaller administrative units
(Eurostat, 2022). Due to its relevance for determining regional policy, data dissemination, and socioeconomic analyses in the
EU, we use this classification as our principal unit of analysis. This further enables direct comparison with the HANZE flood
impact catalogue, which contains data on 2037 reported floods in the study area since 1950, including footprints defined at
NUTS3 level (Paprotny et al., 2023). HANZE also includes exposure and other subnational statistics at the same resolution
(Paprotny and Mengel, 2023). The generation of a high-resolution boundary map of 1422 NUTS3 regions, version 2010, or
their equivalents, is described in Paprotny and Mengel (2023). We further aggregate flood events at national level for
comparison with reported impacts, as this is the typical resolution in which such information is provided. Consequently, the
catalogue is not specific for river catchments or sea basins (as in e.g. Diederen et al., 2019), but for countries and their
subdivisions.
It should be highlighted that the catalogue represents possible floods without considering structural flood protection measures,
hence they are not included in the potential flood footprint estimates. Due to the very limited information on present or past
protection standards, adding estimates of those would potentially create large inaccuracies by filtering out events that happened
in history.





## 2.2 Coastal model

### 2.2.1 Climate data

We model storm surge heights driven by hourly 10-m wind speeds (*u* and *v* component) and surface air pressure, drawing data from the latest ERA5 climate reanalysis (Hersbach et al., 2020). The data were downloaded at a resolution of 0.25° (approximately 28 km at the equator) and then interpolated using first-order conservative remapping (Jones, 1999) to a 0.11° rotated-pole (12.5 km) grid used in our storm surge model, which in turn is the same as the CORDEX grid used in European climate projections (Jacob et al., 2014). Apart from the interpolation, no further adjustments were made to the data.

### 2.2.2 Sea level estimation

The principal component of extreme sea levels are storm surges, which we estimate through a continuous simulation in Delft3D. This hydrodynamic model is commonly applied in continental- or global-scale surge modelling (e.g. Vousdoukas et al., 2016a, Ganguli et al., 2020, Muis et al., 2020). The model set-up is the same as described in Paprotny et al. (2016, 2019), with the difference that it is forced by wind and atmospheric pressure fields from ERA5 instead of ERA-Interim. We also carried out a calibration, using the previous calibration as the starting point, by adjusting the sea bottom roughness coefficients for different basins around Europe, and comparing the modelled surge heights with tide gauge observations for years 2011–2019. This recalibration also benefited from much better availability of observational data, which are described in section 2.5, as they are also used to validate the final simulation. Additionally, the timestep of the model was reduced to 15 min, with outputs saved hourly, compared to 30 min and 6 hours, respectively, in the original version. The model was run from 1 January 1949, with the first year used only as spin-up. Actual ERA5 data was used in the spin-up phase thanks to recent extension of the dataset to 1940.

As storm surge heights are only one component of extreme sea levels, the hourly total water level (*L*) is the combination of six components:

$$L = S + T + W + D + M + G \tag{1}$$

Where:

- *S* is the hourly storm surge height;
- *T* is the hourly tide elevation, computed with pyTMD package (https://github.com/tsutterley/pyTMD) from 34 tidal constituents;
- *W* is the hourly wave run-up, assumed to be 20% of significant wave height (recommended by U.S. Army Corps of Engineers, 2002, used e.g. in Vousdoukas et al., 2016b);
- *D* is the mean dynamic topography defined as the average sea surface height for 1993–2012 above geoid;
- *M* is the long-term variation in sea level related to climatic variation ("sea level rise", SLR), defined as average annual difference from average sea level in year 2000;
- *G* is the glacial isostatic adjustment (GIA) computed from long-term historical rate of change.



Each component was derived from a different source, as summarised in Table 2.

**Table 2. Source of data for computing hourly total water level. * coarser global data were used for northernmost coasts of Europe**

| Component | Source | Spatial resolution | Reference |
|---|---|---|---|
| Storm surge height | Delft3D simulation (this study) | 12.5 km | Paprotny et al. (2016) |
| Tide elevation | FES2014 | 1/16° | Lyard et al. (2021) |
| Wave run-up | ERA5 | 1/2° | Hersbach et al. (2020) |
| Mean dynamic topography | Global Ocean Mean Dynamic Topography (combines global CNES-CLS18 and CMEMS2020 for Black and Mediterranean seas) | 1/8° | Mulet et al. (2021) |
| Sea level rise | 1950–99: Hourly Coastal water levels with Counterfactual (HCC) | 10 km | Treu et al. (2023) |
|  | 2000–2020: European Seas Gridded L4 Sea Surface Heights* | 1/8° | Taburet et al. (2019) |
|  | 2000–2020: Global Ocean Gridded L4 Sea Surface Heights* | 1/4° | Pujol et al. (2016) |
| Glacial isostatic adjustment | ICE-6G_C | 1/5° | Argus et al. (2014), Peltier et al. (2015) |


### 2.2.3 Extracting coastal flood events

As the resolution of each dataset that is used to derive the total water level varies, we assign the nearest grid point of each
model to 5884 coastal segments defined in the coastal flood hazard model (Vousdoukas et al., 2016b) with a nearest-neighbour
approach. From the detrended (1950–2020) hourly timeseries, occurrences of water level above the 99.6[th] percentile were
identified and considered potential coastal floods. Occurrence of water levels below the 99.6[th] percentile for at least two full
calendar days separated two events from each other. Such thresholds lead to, on average, about five potential flood events per
year. Then, events were aggregated according to NUTS3 regional boundaries, again with the principle that the beginning of
any segment-level flood event in a NUTS3 region has to occur at least two full calendar days after the end of any previous
segment-level event in that region.

### 2.2.4 Deriving coastal flood footprints

For each coastal segment in the dataset, an extreme value analysis was carried out using a Generalised Pareto distribution and
a peak-over-threshold approach. This enabled deriving extreme sea level scenarios (return periods of 2, 5, 10, 20, 30, 50, 100,
200, and 500 years) for coastal inundation modelling. This was carried out according to a methodology developed by
Vousdoukas et al. (2016b). Briefly, the maps were generated with the Lisflood-ACC (LFP) model (Bates et al., 2010) applied
at 30 m spatial resolution. In terms of Digital Elevation Model (DEM), we use the recently published GLO-30 DEM (European
Space Agency and Sinergise, 2021) after applying post-processing using global LIDAR observations to further remove vertical
bias, correcting for buildings and vegetation. The description of the GLO-30 post-processing is described in detail in Pronk et
al. (2023). The simulations consider gridded hydraulic roughness values derived from land-use maps (Zanaga et al., 2021).



Lisflood-ACC is applied for each coastal segment with the model domain extending up to 200 km landwards in order to ensure
the inclusion of all potentially hydrologically connected areas that may lie inland and away from the coast.
Total water level of each segment-level flood event is linked with the water level used to generate the flood hazard maps for
each segment. In this way, it is possible to interpolate water depths from the stack of hazard maps to event-specific extreme
sea levels. This is only done if the water levels for an event exceed a flood threshold, defined as the higher of the two following
thresholds:
● Total water level with a 2-year return period, derived from the Generalised Pareto distribution;
● Maximum observed total water level minus storm surge height.
The first threshold was chosen for consistency with the riverine model as it is akin to the typical definition of a bank-full river
discharge. The second threshold was added to avoid overestimating risk in regions (mainly Eastern Mediterranean), where
storm surge heights are very low, but wave run-up contributes significantly to extreme sea level.
Only grid cells with water depths of at least 10 cm were considered inundated for consistency with riverine flood maps. The
individual flood maps for each coastal segment were aggregated within a NUTS3-level event. Finally, only those NUTS3-
level events were preserved for further analysis if the potential flood zone was at least 100 ha. As further processing is carried
out together with the riverine model, we now describe the river component, and continue explaining the next steps towards the
combined flood catalogue in section 2.4.
**2.3 Riverine model**
**2.3.1 Climate data**
We used river discharge from Tilloy et al. (2024) that was modelled using ERA5-Land, which is a downscaled version of
ERA5 characterised by 0.1° (approximately 11 km at the equator) resolution (Muñoz-Sabater et al., 2021). It was further
statistically downscaled and bias adjusted to 1' (arc minute) resolution using ISIMIP3BASD v3.0.0 method developed by
Lange (2019, 2022), using EMO-1 gridded observational data, which is a 1' variant of the EMO-5 dataset developed by
Thiemig et al. (2022). Temperature and precipitation with 6-hourly resolution were used as the primary driver of the
hydrological model, while potential evapotranspiration was computed at daily resolution using the LISVAP model by van der
Knijff (2006). For details on the preparation of the meteorological data, we refer to Tilloy et al. (2024).
**2.3.2 River discharge simulation**
River discharges were modelled through continuous simulations using the LISFLOOD hydrological model (Burek et al., 2013)
implemented in the European Flood Awareness System, or EFAS (Copernicus Emergency Management Service, 2023). Tilloy
et al. (2024) used the latest model set-up, v5.0 (Choulga et al., 2023), and simulated river discharges with meteorological
inputs described in section 2.3.1. The EFAS model was run starting 3 January 1950 following the 71-year pre-run. Due to
rapid evolution of socioeconomic conditions in the catchments of Europe, the input socioeconomic maps were changed with



the start of every new calendar year of the simulation. The evolving socioeconomic conditions included land use (in six
classes), reservoirs (based on the year of construction of each dam), and water demand (in four sectors). For details on the
river discharge simulation and its validation, we again refer to Tilloy et al. (2024).

### 2.3.3     Extracting riverine flood events

The output of the river model is a time series of 6-hourly discharge for 7.5 million grid cells. Due to the availability of flood
hazard maps for footprint estimation (section 2.3.4), we extract data only for 282,528 grid cells that have an upstream area of
at least 100 km². Occurrences of discharge above the 98th percentile (on annual basis) were identified and considered potential
riverine floods. Occurrence of water levels below the 98th percentile for at least two full calendar days separated two events
from each other. As in the coastal model (section 2.2.3), those thresholds were intended to produce roughly five potential flood
events per year in each grid cell. Then, events were aggregated according to NUTS3 regional boundaries, again with the
principle that the beginning of any grid cell-level flood event in a NUTS3 region has to occur at least two full calendar days
after the end of any previous grid cell-level event in that region.

### 2.3.4 Deriving riverine and compound flood footprints

For each grid cell in the dataset, an extreme value analysis was carried out using a Generalised Pareto distribution and a peak-
over-threshold approach, where the peak discharge was detrended based on annual maximum discharge for 1950–2020. In
contrast to the coastal model (section 2.2.4), no additional hydrodynamic modelling was carried out in the riverine model.
Instead, the flooding processes were represented using the dataset of flood hazard maps developed by Dottori et al. (2022),
which are available for a range of return periods from 10 to 500 years for grid cells with an upstream area above 500 km² . The
maps were generated with the Lisflood-ACC (LFP) model (Bates et al., 2010), applied at 100 m spatial resolution and driven
by hydrological simulations from a previous set-up of EFAS (Arnal et al., 2019). In this study, given the different resolutions
of the LISFLOOD simulations and the flood hazard maps, the two datasets were matched according to the procedure described
in Dottori et al. (2022).
To provide coverage for smaller catchments, the flood maps by Paprotny et al. (2017) were applied for grid cells with an
upstream area of 100–499 km². The maps for five scenarios (return periods of 10, 30, 100, 300 and 1000 years) were based on
discharges estimated with a Bayesian Network-based model from Paprotny and Morales-Nápoles (2017). The simulations
were performed using a one-dimensional 'steady-state' hydraulic model Deltares SOBEK to obtain water levels along rivers.
Those levels were then used to generate water depth maps over a digital elevation model. The maps use the exact same grid as
the ones from Dottori et al. (2022). For details on the methodology and validation of the maps we refer to Paprotny et al.
219   (2017).

Peak river discharge per each grid cell during a given potential river flood event was linked with the scenarios used to generate
the flood hazard maps so that the appropriate maps were used to interpolate water depths. If the return period of the peak
discharge was below 10 years, water depths were extrapolated using two maps with the lowest return periods. No flooding was



assumed if the peak discharge was below the empirical 2-year return period derived from detrended 1950–2020 peak discharges
of the extracted flood events. This threshold was typically much lower than the 2-year return period derived with the
Generalized Pareto distribution.
Only grid cells with water depths of at least 10 cm were considered inundated, as in the maps of Dottori et al. (2022). The
individual flood maps for each river grid cell were aggregated within a NUTS3-level event. Finally, only those NUTS3-level
events were preserved for further analysis if the potential flood zone was at least 100 ha. At this point, the list of NUTS3-level
events was compared against the same list from the coastal model. If a river event in a given NUTS3 region occurred at the
same time as a coastal event in the same region, a separate "compound" event was created by merging the flood zones of the
coastal and riverine events in that region. The compound events are analysed in addition to the individual coastal and riverine
events, rather than replacing them. From here, processing of the potential flood events follows a common path for all types of
events.
**2.4 Combined flood catalogue**
**2.4.1 Aggregating and estimating potential losses per event**
Almost 250,000 potential flood events at the level of NUTS3 regions are aggregated for each country. One full calendar day
separates two country-level events consisting of at least one NUTS3 event. Coastal, riverine, and compound events are each
aggregated separately. Each event is characterised by hydrological parameters, such as inundated area, average water depth,
duration and return period. The latter is the geometric average of all river grid cells or coastal segments that contribute to the
flooded area.
Potential losses were estimated by multiplying exposure for each 100 m grid cell within each flood footprint with an appropriate
loss function. Exposure per grid cell (population and value of fixed assets) was computed with the HANZE v2.0 exposure
model (Paprotny and Mengel, 2023), which estimates historical exposure changes using a combination of rule-based and
statistical modelling that enabled downscaling past demographic and economic trends at subnational level into a high-
resolution grid. The model provides annual data for years 2000–2020 and 5-yearly timesteps for 1950–2000. Alongside
population, the model can generate values of tangible fixed asset stock in euros (constant 2020 prices and exchange rates) in
8 sectors (housing, consumer durables, agriculture, forestry, industry, mining, services, infrastructure).
Firstly, fatalities were estimated per each 100 m grid cell by multiplying the population with the death probability determined
by water depth. Due to the lack of velocity data or dike breach locations, only such a simplified approach can be used here.
We opted for the S-shaped depth-fatality function by Boyd et al. (2005) as presented in Jonkman et al. (2008), which shows
very low chance of death until water depths of approximately 3 m, i.e.:
$$F_D = \frac{0.34}{1 + exp(20.37 - 6.18d)} \qquad (2)$$

where $F_D$ is the mortality rate and $d$ is the water depth in m.



The second indicator, people affected, is simply the total population within the flood footprint. Finally, economic losses were estimated using a set of depth-damage functions for different economic sectors. We applied the logarithmic-type functions proposed for Europe by Huizinga et al. (2017) that distinguish five sectors: agriculture, industry, commercial, infrastructure, and residential. The functions were applied to the appropriate sector in the exposure model. It should be noted that whenever "economic losses" are mentioned in this paper, they only refer to direct damage to tangible fixed assets, without considering indirect impacts.

**2.4.2 Obtaining the final flood catalogue**

Estimated flood impacts of each event computed in the previous step were used to further filter the flood event catalogue only to those floods with significant potential for socioeconomic impacts. To qualify for the list, the event had to pass two thresholds simultaneously (Table 3):

- Inundated area above a fixed threshold, and
- At least one of two socioeconomic impact indicators (computed according to section 2.4.1):
  - people potentially affected above fixed threshold, or
  - potential economic losses above an event-specific threshold.

The exact threshold depends on the type of event, and in case of economic losses also on country and year of event, as it was linked to the level of gross domestic product (GDP) per capita (Table 3).

**Table 3. Thresholds for selecting flood events with significant potential impacts.**

| Threshold | Coastal floods | Riverine and compound floods |
|---|---|---|
| Area inundated | 1000 ha | 2000 ha |
| People affected | 2500 | 5000 |
| Economic damage | 10,000 times GDP per capita (country and year of event) | 20,000 times GDP per capita (country and year of event) |

Thresholds in Table 3, as well as those described earlier in the methodology, were selected iteratively based on the following objectives:

- Maximise the number of modelled events matching observed events from HANZE;
- Maximise the share of one-to-one relationships between modelled and observed events (as opposed to many-to-one or one-to-many relationships);
- Minimise the spatial extent of events in terms of affected NUTS3 regions beyond those indicated in HANZE;
- Create a list of events large enough for statistical analyses and small enough to allow manual searches of historical records for all events.



281 To help select the thresholds, observed flood events from the following six datasets were matched per country according to

282 start and end dates:

- HANZE v2.1 (Paprotny et al., 2023);
- EM-DAT (Centre for Research on the Epidemiology of Disasters 2023);
- EEA Flood Phenomena (from 1980 only) (European Environment Agency, 2015);
- Dartmouth Flood Observatory (from 1985 only) (Brakenridge, 2023);
- FFEM-DB (from 1980 only) (Papagiannaki et al., 2022);
- Recorded Flood Outlines (England only) (Environment Agency, 2023).

In addition, the HANZE dataset was matched with events below the tested thresholds .Following the above objectives results
in different potential impact thresholds for coastal and riverine floods.

### 2.4.3 Comparing modelled and reported events

The modelled flood events of the catalogue were evaluated using gauge records and impact data as well as manual research
involving all kinds of documentary sources. At first, English-language papers and local-language flood catalogues providing
an overview of the hazard in the country were consulted. Then, national disaster databases were searched and the relevant data
was extracted. Papers on case studies of disasters were searched for in both English and the local language of the country being
researched. A keyword-based search in both English and the local language was performed using a web engine to identify
news articles or other online reports mentioning the relevant disasters. In total, 946 major text or data sources were used, 828
of which are listed in the HANZE v2.1 dataset (Paprotny et al., 2023) and the remainder is listed together with the data from
this study. Based on this information on impacts, each event was categorised into one of the classes listed in Table 4.

**Table 4. Classification of flood events considering the availability of data sources as well as reported hydrological and socioeconomic**
**impacts.**

| Class | Short name | Evaluation result | |
|---|---|---|---|
| | | Extreme hydrological event | Inundation with significant socioeconomic impacts |
| A | Impacts, data | Confirmed by sources | Confirmed by sources (impact data available) |
| B | Impacts, no data | Confirmed by sources | Confirmed by sources (impact data not available) |
| C | No impacts | Confirmed by sources | Not confirmed by sources |
| D | Unknown impacts | Confirmed by sources | No sources available |
| E | False positive | Not confirmed by sources | Not confirmed by sources or no sources available |
| F | No information | No sources available | No sources available |


In applying the classification from Table 4, a decision graph from Fig. 1 was used. In general, in case of complete lack of
gauge data or documentary sources, the event was labelled F ("No information"), meaning that no observational data is
available and therefore modelled data can be neither confirmed or rejected. In case gauge records are available, it was firstly
evaluated if they indicate extreme values. Exceedance of a 2-year return period was considered sufficient to confirm that the
modelled event was an extreme hydrological event in real life. If the threshold was not exceeded at any of the available gauge



stations, the time series was analysed, and the event was considered confirmed as hydrologically extreme if a flood wave was
clearly visible at the dates indicated by the model. If no flood wave was visible, the event was considered a "False positive"
(label E), i.e. an error of the model that indicates a too high simulated river discharge or sea level. In rare cases, this
classification was overridden if documentary sources indicated the occurrence of a flood event.
For events confirmed as hydrologically extreme, further analysis concentrated on the occurrence of significant socioeconomic
impacts. Here, significant impacts were defined as in the HANZE database (Paprotny et al., 2023), i.e. exceedance of at least
one the following thresholds:

316        ● At least 1000 ha (10 km²) inundated;

317        ● At least one person killed or missing presumed dead;

318        ● At least 50 households or 200 people affected by their homes being inundated or who were evacuated;

319        ● Losses in monetary terms corresponding to at least 1 million euro in 2020 prices.

In case no further information was available, the event was labelled D ("Unknown impacts"). If despite good coverage of
sources (e.g. comprehensive local/national flood databases or catalogues), no impacts are mentioned, or in rare cases, direct
statement that e.g. a flood emergency did not result in breaching of flood defences, the event was labelled C ("No impacts").
Also, if data on impacts were available, but they did not pass any of the aforementioned thresholds, the event was labelled as
"No impacts". Events with sufficient information on significant impacts were labelled A ("Impact, data") and incorporated
into the HANZE database. However, if statistical data was not accessible, or referred only to a small part of the impacted area,
but available descriptions strongly indicated that one of the impact thresholds was likely exceeded, the event was recorded in
a separate list of events, labelled B ("Impact, no data"). Available historical information was collected for such an event in a
database that is a simplified version of HANZE. Detailed description of the data collected in this database, which is made
publicly available with this study, is provided in Appendix A1. It should be noted that a matching of dates and country with
historical events was not enough to label the event A or B. For that, at least one NUTS3 region affected during the event had
to be correctly identified by the model.




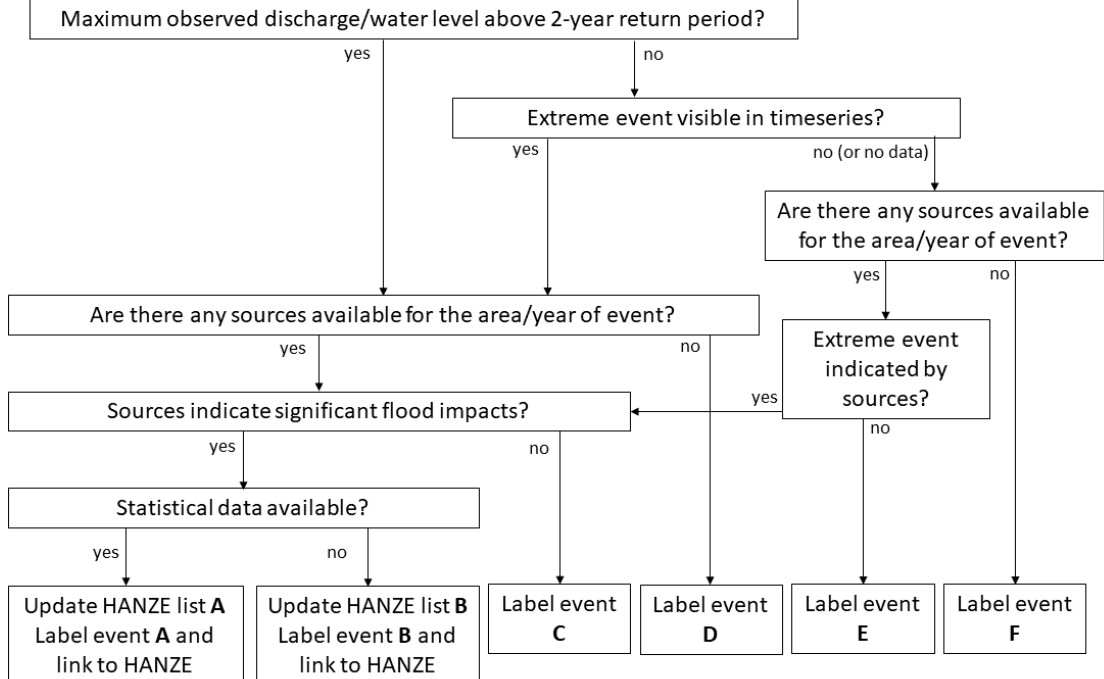

**Figure 1. Decision graph for classifying flood events.**

The final flood catalogue consists of two components: (1) a table with all events, indicating their timing, location, potential
impacts, hydrological parameters and classification, and (2) potential flood footprint maps in vector format. The data contained
in the table are explained in Appendix A2.
**2.5 Validation**
Validation of river discharges is presented by Tilloy et al. (2024), however we used the 3442 stations containing daily
observations collected for that study for further analysis. The dataset helped us to classify the events in section 2.4.3. Further,
we compared extreme discharges observed during riverine and compound events with modelled discharges. Station data was
obtained in 60% from the Global Runoff Data Centre and in 40% from national public datasets of France, Norway, Poland,
Spain, Sweden and the United Kingdom. The analysis was limited to 2914 stations with an upstream area of at least 100 km²,
located in the affected NUTS3 regions according to the model. If the event duration and available gauge series were both at
least 30 days, the daily discharge was compared using the Kling–Gupta efficiency, or KGE (Gupta et al., 2009), and
Spearman's coefficient of determination (as Pearson's is used in the KGE score). Otherwise, an equal amount of days was
added before and after the event, so that at least 30 observations are used. The maximum daily discharges during the event
were also compared.





Validation of the hourly storm surge heights, tide elevations and combined water level was done using 428 tide gauges. Almost
all stations (413) were gathered from GESLA v3 dataset (Haigh et al., 2023), but for better coverage of the eastern
Mediterranean Sea it was complemented with 7 stations from Poseidon System (2023), and for the southern Baltic Sea with 8
stations from the Institute of Meteorology and Water Management – National Research Institute (2023). Apart from validation
for all available time series, an event-based validation was done as for river discharges. The default time window for the
comparison between modelled and observed data was 7 days, unless the event had a longer duration.
Finally, the modelled flood footprints were compared with satellite-derived footprints from the Global Flood Database (GFD,
Tellman et al., 2021). The footprints were converted into vector layers, with permanent water bodies removed from them, as
per data contained in GFD. Only footprints within NUTS3 regions indicated as affected in the HANZE database were included
in the analysis. Population affected within the footprints was derived from HANZE population maps. Flooded area and
population affected based on footprints from this study and GFD were compared with reported impacts. Additionally, all flood
events in the catalogue with comparative reported impact data were analysed for the difference in modelled and reported
impacts. Ideally, all modelled impacts should be higher than what was reported, as the intention of the catalogue is to generate
potential footprints that do not consider flood protection. Finally, footprints from this study and GFD were intersected to derive
the hit rate, i.e. share of the satellite footprints correctly reproduced by the model. This is a similar approach that was used to
validate flood hazard maps that are the basis of the modelled footprints (Vousdoukas et al., 2016b, Paprotny et al., 2017,
Dottori et al., 2022).

**3 Results**

**3.1 Flood event catalogue**

**3.1.1 Modelled impacts by classification**

The final catalogue includes 2436 coastal, 11,205 riverine, and 1058 compound events with significant potential for
socioeconomic impacts (Fig. 2). This already indicates a significant proportion of coastal and riverine events might be
compound events. The spatial location and timeframe of events was matched with at least some gauge observations for 63%
of coastal and 72% of riverine events. By applying the 2-year return period threshold to observational data, it was possible to
immediately confirm that 40% of coastal and 45% of riverine events were hydrologically extreme. Further confirmations were
obtained through analysis of gauge timeseries and documentary records, increasing the confirmation rate to 80% for coastal,
77% for riverine, and 66% for compound events. On the other hand, no extreme event was indicated by gauge or documentary
sources for a small part of the catalogue. The false positive ratio ("E" events to "A"-"D" events) amounts to only 2.2% for
compound, 3.3% for coastal, and 5.2% for riverine floods.




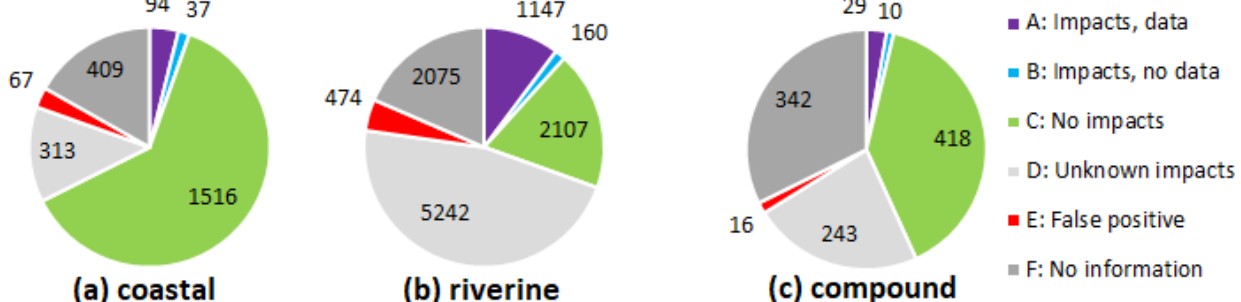


**Figure 2. Flood events in the catalogue by classification: (a) coastal, (b) riverine, and (c) compound.**

Confirmation, or at least high confidence based on available documentary sources, whether the event did, or did not, result in
significant socioeconomic impacts was possible for the majority of coastal and compound events, but not for riverine floods.
However, the latter occurred by far most frequently, and it was possible to confirm significant socioeconomic impacts for
11.7% of riverine, 5.4% of coastal, and 3.7% of compound events (Fig. 2). In some cases, "A" ("Impacts, data") events
correspond to more than one reported flood in the HANZE database, or the events are a combination of "A" and "B" ("Impacts,
no data")-type events. Therefore, the 1270 "A" and 207 "B" events actually correspond to 1471 historical floods in HANZE
and 237 historical floods without impact data collected in a separate dataset as part of this study (see Appendix A1). This
statistic excludes a small number of events that were below the significant impact threshold, but indicated a temporal match
with the HANZE database. Only 109 such events were identified, of which only two were coastal events and two were
compound events. Out of those, only 33 events, all riverine, were spatially matched with HANZE, a single historical flood in
each case. This constitutes only 2% of matched HANZE events, hence we can deem the hydrological and socioeconomic
thresholds in this study as well designed, as few HANZE events were missed due to their imposition without creating too many
non-impact events. Also, while there were many one-to-many matches between our model and HANZE, largely due to the
data-availability rules causing splitting of some flood events in HANZE, there were only a handful of cases of many-to-one
connections.
The distribution of events over time (Fig. 3) shows an upward trend, which in case of "A" and "B" events is largely related to
better availability of data. There is also better confidence in non-occurrence of impacts for coastal and compound events in
recent decades compared to the beginning of the timeseries. An increase in "F" events in the final few years for riverine and
compound events is primarily connected will lower availability of recent river gauge data.



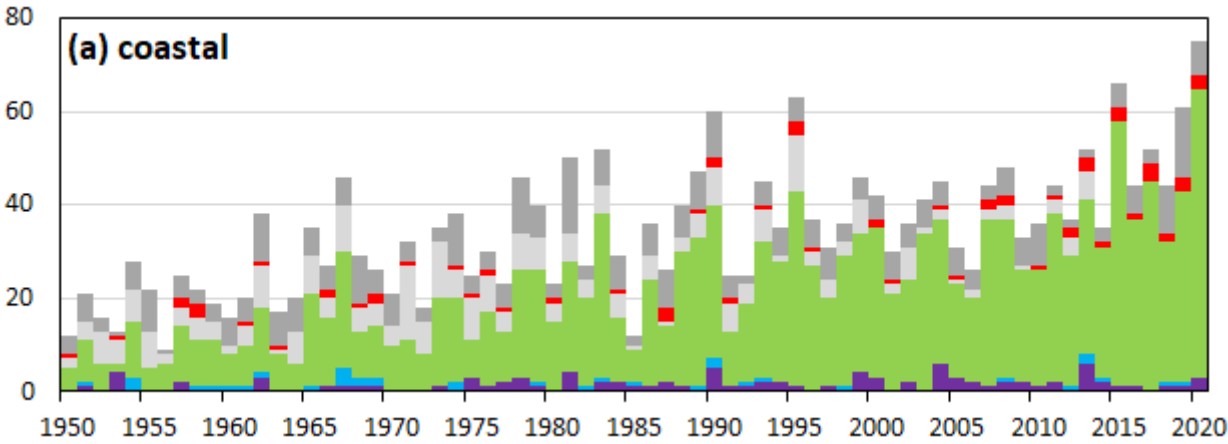



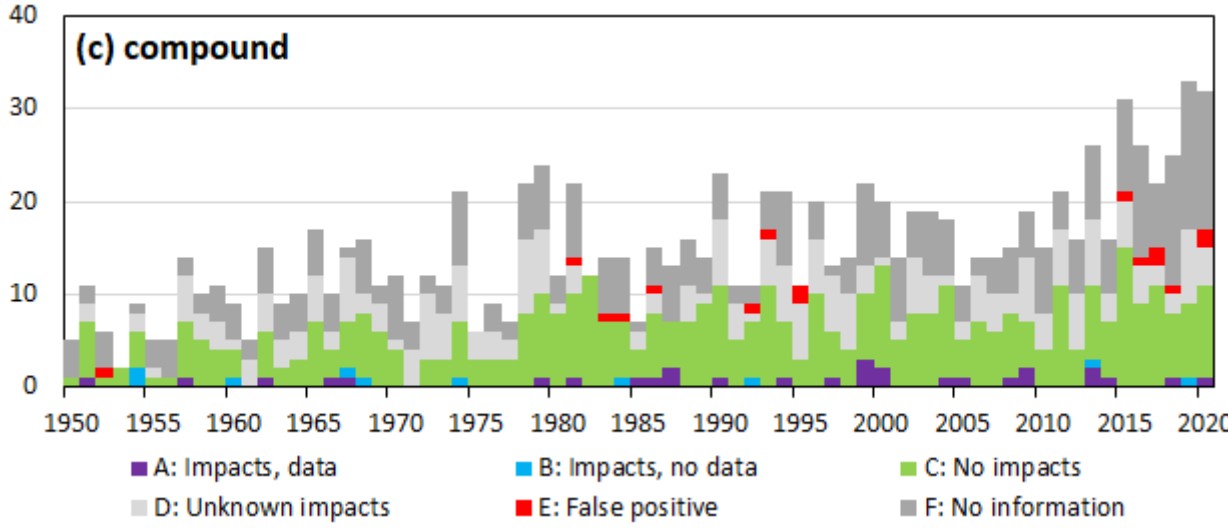


**Figure 3. Flood events in the catalogue by year and classification: (a) coastal, (b) riverine, and (c) compound.**

Modelled extremity and impacts of events vary strongly by class (Fig. 4). The return period along affected river and coastal
segments is generally much higher for "A" and "B" events compared to all others. 18% of coastal and 37% of riverine events,
in which the geometric average of return periods in the affected area was above 25 years, was classified as either "A" or "B".
In contrast, when the return period was below 5 years, the values were 2% and 10%, respectively. Interestingly, the occurrence
of "F" class ("No information") was only slightly lower for higher return periods. Confirmed impactful events were also longer
in duration than other classes, with false positives ("E") having the shortest duration. Consequently, the "A" and "B" events
had, on average, the highest impact potential. In Fig. 4c, the dimensionless damage index is the average of four impact
categories (potential area inundated, fatalities, persons affected, and economic loss) relative to maximum impact of any event
in the country during 1950–2020 at constant 1950 exposure. False positives had, on average, the lowest impact potential. In
all examples, the remaining categories ("C" – No impacts, "D" – Unknown impacts, and "F") oscillated around the average
values for all variables analysed in Fig. 4.

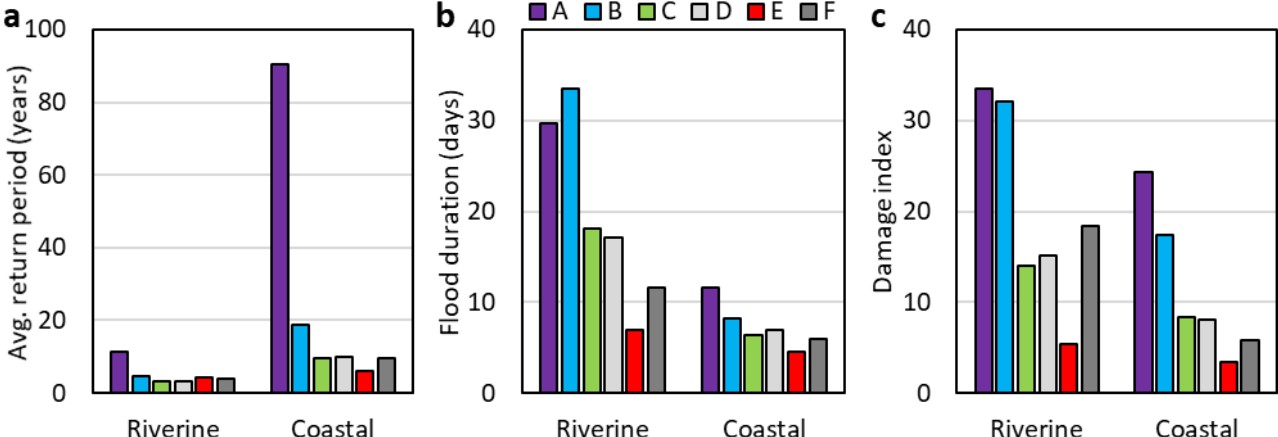


**Figure 4. Comparison of mean values of selected indicators by main flood type and classification: (a) average return period along**
**affected river or coastal segments, (b) total flood event duration, (c) dimensionless damage index, where 100 equals the highest**
**potential impact of any event in the country during 1950–2020 at constant exposure.**
**3.1.2 Comparison with HANZE reported impacts database**
The flood catalogue includes the majority of reported historical floods with significant socioeconomic impacts since 1950
contained in the HANZE v2.1 database (Paprotny et al., 2023). However, there is a strong difference between the completeness
of the catalogue according to flood type. While about 90% of coastal, compound and slow-onset riverine floods were modelled,
only 55% of flash floods were captured (Table 5). The latter category, as defined in HANZE, represents short, rapid floods,
where the extreme rainfall event triggering the event lasted no more than 24 hours, excluding urban floods. As those often





occur in small catchments, they are often not captured as the study was limited only to catchments with an upstream area of at
least 100 km$^2$.
The HANZE database indicates more than 6,000 NUTS3-level impacts since 1950. 78% of those are reproduced by the model
(Table 5), a slightly higher percentage than the hit rate at event level (74%). This is largely due to good coverage for slow-
onset riverine floods (88%) compared to flash floods (55%), when the former affected more regions on average than the latter.
For the 1504 events matched by the model, the hit rate of NUTS3 regions for the model is 89%, again lower for flash floods
(84%) than for larger riverine events (91%), not to mention coastal floods (98%). A full list of HANZE events with the
information which of those were captured by the model, and which NUTS3 regions were correctly identified is provided
together with the dataset on the repository (Paprotny, 2024). In general, performance of the model is stable over time (Fig. 5),
though the share of events correctly identified by the model is lower in the very beginning of the model runs (1950s).
Analysing the reported impacts in HANZE, even though they are incomplete (except for fatalities), provides further insights.
The data in Table 5 show that 97–100% of reported impacts in all four categories for coastal, compound and slow-onset riverine
floods were in those historical floods that could also be found in the model. This shows that the model captured almost all
large events, and the omissions are mostly minor floods in specific areas where the hazard is apparently not well quantified.
For instance, out of 14 omitted coastal and compound floods, 10 are events in Italy occurring mostly before 1964 and affecting
200–500 persons with no more than one fatality (with a single exception of a seven-fatality flood from January 1950). Much
lower coverage is again for flash floods, as only those responsible for 61% of all fatalities can be found in the model. For other
impact categories, the coverage is better, but historical records are very incomplete in relation to those statistics.






**Table 5. Comparison of the number of HANZE events, their footprints and reported impacts, with modelled data, 1950–2020. * only regions classified as compound by the model – regions forming compound events in the HANZE database are not necessarily in the zone directly influenced by both riverine and coastal drivers; ** impact data is not available for all HANZE events.**

| Category | HANZE event type | | | | All events |
| --- | --- | --- | --- | --- | --- |
| | Coastal | River/ Coastal | River | Flash | |
| *Matching of events with impact data ("A" events)* | | | | | |
| Number of events in HANZE database (1950–2020) | 71 | 41 | 970 | 955 | 2037 |
| Number of modelled events matched with HANZE | 61 | 37 | 880 | 526 | 1504 |
| Percentage of HANZE events matched with modelled events | 90% | 86% | 91% | 55% | 74% |
| *Matching of affected NUTS3 regions* | | | | | |
| Number of affected NUTS3 regions in HANZE database | 195 | 162 | 4058 | 1671 | 6086 |
| Number of affected NUTS3 regions in matched HANZE events | 180 | 152 | 3910 | 1084 | 5326 |
| Number of regions that are also in the modelled events | 177 | 97* | 3553 | 915 | 4742 |
| Percentage of all regions that are also in the modelled events | 91% | 60%* | 88% | 55% | 78% |
| Percentage of matched regions that are also in the modelled events | 98% | 64%* | 91% | 84% | 89% |
| *Percentage of total reported impacts of all HANZE events within matched HANZE events (1950–2020) ** * | | | | | |
| Area inundated | 99.8% | 100% | 99.5% | 93.2% | 99.2% |
| Fatalities | 99.5% | 99.4% | 97.0% | 61.2% | 81.2% |
| Persons affected | 99.3% | 98.7% | 98.9% | 78.9% | 96.3% |
| Economic losses in 2020 euros | 99.8% | 100% | 98.9% | 86.1% | 96.1% |
| *Matching of events without impact data ("B")* | | | | | |
| Number of historical floods without impact data (list B) | 27 | 12 | 119 | 79 | 237 |



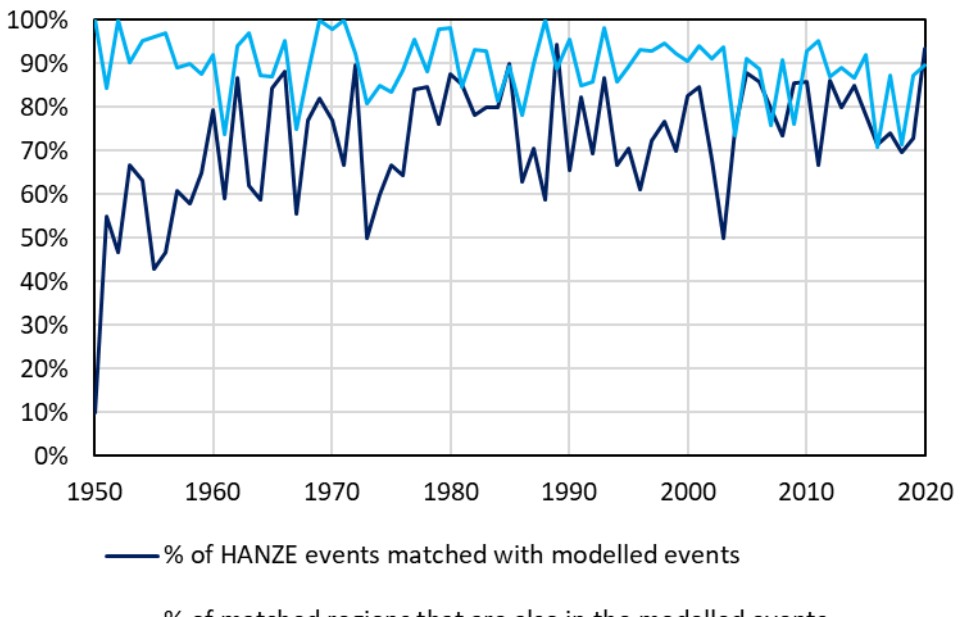

**Figure 5. Share of HANZE events matched with the model, and the share of regions in matched events also present in the model.**

## 3.2 Modelled potential impacts in the flood catalogue

Without flood protection measures, floods would have large consequences throughout Europe. A simple summation of flood impacts in the catalogue is not informative, as it assumes not only no flood protection, but also that population and economic activity move into the frequently affected zone in the first place, and then immediately return to previous conditions after each event, even just days after the previous. Considering the total reported impacts in HANZE v2.1, albeit incomplete, it can be estimated that only about 1% of potentially inundated area, population and economic assets were actually affected during 1950–2020. The reported flood deaths equal only about 0.01% of the potential fatalities. Therefore, the potential impacts are merely an intermediate result necessary in the process of estimating flood vulnerability and impact attribution (see section 5). Still, some analysis of the results can be performed as the modelling chain can derive the impact estimates under different exposure scenarios, and it was driven by variable climate conditions.

### 3.2.1 Temporal changes in potential flood impacts

For all types of events, an increase in the number of potential events and their impacts was recorded (Table 6). Even though the trends are less pronounced under constant exposure scenarios, they are still equivalent to at least 21% increase in potential coastal flood losses in an average year between 1950 and 2020 in case of fatalities, 47% in case of economic loss and 75% in case of affected population. For riverine floods, the potential impacts have grown even more, while the strongest increase is



indicated for compound floods, at least threefold since 1950. Potential impacts per flood event are rather similar for coastal
and riverine events, and slightly lower for compound events, as the latter category is spatially constrained to regions directly
affected by both coastal and riverine drivers.
Demographic and economic growth since 1950 has increased potential losses substantially. Presently, exposure of population
to riverine floods is more than 50% higher than if population would have not increased, and nearly twice as high for coastal
and compound events. Potential impacts relative to the total population in the study area increase more strongly than in the
constant-exposure scenario, indicating stronger population growth in areas prone to coastal and compound flooding relative to
areas not at risk. However, only a marginal increase in areas at risk of riverine floods was observed relative to areas not prone
to this type of floods.
Enormous increase in gross domestic product (GDP) per capita (2% per year in the study area), and associated growth in the
stock of fixed assets resulted in a five- to six-fold increase in potential losses relative to 1950, and eight- to ten-fold increase
in 2020. As the asset growth was higher than GDP, potential economic losses relative to GDP also increased between 1950
and 2020. In contrast to population growth, asset growth in flood-prone areas was only marginally higher, or even lower in
case of riverine events, than in areas not at risk of flooding.




**Table 6. Average potential impacts of floods and their trends, by flood type and exposure scenario (dynamic year-of-event exposure, or fixed at 1950 or 2020 levels). The impacts of compound events mostly overlap with those of coastal and riverine, therefore they should not be added together. Economic losses in constant 2020 prices and exchange rates.**

| Flood type | Coastal | | | Riverine | | | Compound | | |
|---|---|---|---|---|---|---|---|---|---|
| Exposure map | Dyna-mic | 1950 | 2020 | Dyna-mic | 1950 | 2020 | Dyna-mic | 1950 | 2020 |
| *Average potential impacts per year* | | | | | | | | | |
| Number of events | 34 | x | x | 158 | x | x | 15 | x | x |
| Area inundated (thsds. km$^2$) | 27 | x | x | 182 | x | x | 13 | x | x |
| Fatalities (thousands) | 214 | 133 | 351 | 1,059 | 851 | 1,246 | 81 | 51 | 108 |
| Persons affected (thousands) | 2,689 | 1,966 | 3,590 | 15,284 | 11,919 | 18,247 | 1,004 | 704 | 1,239 |
| Economic loss (billion euro) | 237 | 50 | 478 | 1,200 | 261 | 2,196 | 86 | 14 | 149 |
| *Annual increase of potential impacts (%)* | | | | | | | | | |
| Number of events | 1.3 | x | x | 0.7 | x | x | 1.5 | x | x |
| Area inundated | 1.1 | x | x | 0.4 | x | x | 1.6 | x | x |
| Fatalities | 1.5 | 0.4 | 0.3 | 1.0 | 0.6 | 0.6 | 2.6 | 1.6 | 1.9 |
| Persons affected | 1.5 | 0.9 | 0.8 | 1.2 | 0.8 | 0.8 | 2.4 | 1.7 | 1.9 |
| Economic loss | 2.8 | 0.6 | 0.5 | 3.1 | 0.9 | 0.9 | 4.0 | 1.8 | 2.0 |
| *Increase in total impacts relative to 1950 exposure* | | | | | | | | | |
| Fatalities | 61% | x | 164% | 24% | x | 46% | 59% | x | 111% |
| Persons affected | 37% | x | 83% | 28% | x | 53% | 43% | x | 76% |
| Economic loss | 371% | x | 852% | 360% | x | 742% | 505% | x | 948% |

## 3.2.2 Spatial distribution of potential flood impacts

Coastal and compound flood potential is highly concentrated in just a few countries (Fig. 6). Though these estimates do not include the effect of flood protection, the top five countries by coastal flood potential are also most prominently featured in the HANZE database in terms of historical coastal flood impacts: the Netherlands, the United Kingdom, Germany, France, and Italy. The same group, plus Ireland, also have the most significant compound flood potential. On the other hand, numerous potential coastal and compound floods are present in the catalogue for Greece, but only one historical example for that country could be found in HANZE (a compound flood in 1968 that affected Crete).

In total, the flood catalogue includes coastal floods in 25 countries and compound floods in 24. Slovenia also has no event on the compound flood list, as none of the compound events was able to pass the higher socioeconomic thresholds for riverine



and compound events. Bosnia and Herzegovina and Montenegro are the only countries on the compound flood list that are not present on the coastal flood list due to the limited risk along their short coastlines. Bulgaria is the only country with access to the sea that is not included in the coastal flood catalogue, as no event exceeded the socioeconomic thresholds. One historical case of coastal flooding in Bulgaria (in 1999) was recorded in HANZE.

Riverine flood potential is more evenly distributed in space. All countries highlighted in Fig. 6b have numerous examples of historical damaging floods in HANZE, with the exception of the Netherlands, where historical cases are limited to four floods recorded in the 1990s. In total, 37 out of 42 countries in the study area had at least some potential flood events. Some small countries had no riverine or compound floods in the catalogue, as they have no river section with an upstream area bigger than 100 km$^2$.

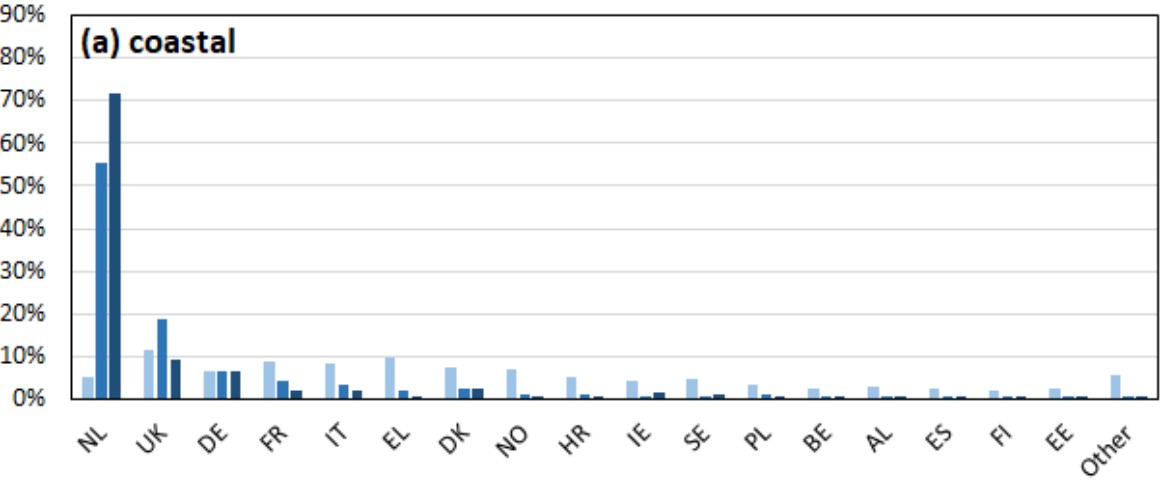

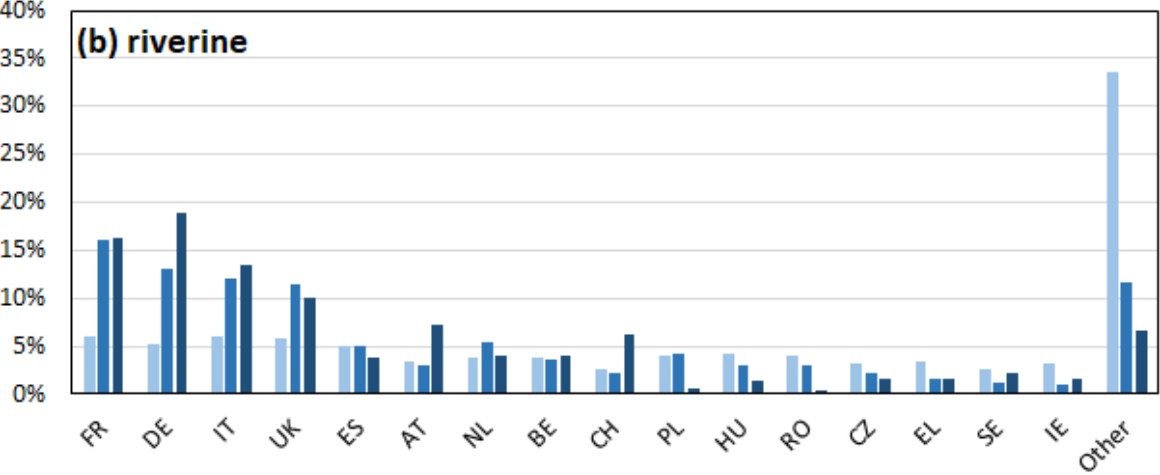



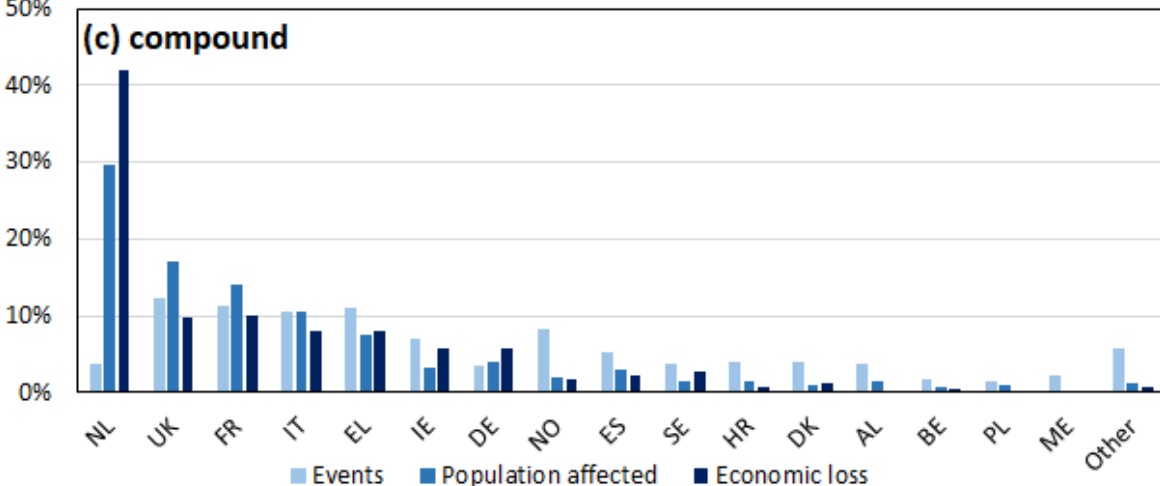

**Figure 6. Flood events in the catalogue by country and potential impacts, as % of all events: (a) coastal, (b) riverine, and (c) compound. Population affected and economic loss in constant 2020 exposure.**

A variety of indicators can be derived at the level of NUTS3 regions. Here we present one example, potential economic damages normalised to 2020 exposure level, relative to 2020 gross domestic product (GDP). Along most of the European coast potential damages resulting from storm surges are limited (Fig. 7), with risk concentrated along the North Sea, Adriatic Sea, and Aegean Sea. Locations of the most significant past coastal floods stand out (the Netherlands, German Bight, Venice). Riverine damage potential is much higher (Fig. 8), and concentrated around main European mountain ranges (Alps, Carpathians, Pyrenees, Dinaric Alps), as well as Scandinavia and British Isles. Risk is noticeably lower along the Northern European Plain, southwestern Iberian Peninsula, and southern Great Britain. However, it must be stressed that the data represent only damage potential, without considering flood protection or other forms of adaptation.

In some parts of Europe, the possibility of co-occurrence of coastal and riverine floods could have large implications on risk. Fig. 9 maps the share of compound flood potential at regional level relative to the total. For each NUTS3 region, we derived a list of all flood events with a potential inundated area of 100 ha, i.e. before aggregation and application of socioeconomic thresholds, then removed riverine and coastal events that overlapped with compound events. In this way, it was possible to avoid double counting and sum together the remaining flood events. The results (Fig. 9) show that compound potential is very unevenly distributed across Europe. In northern and eastern coasts of the Adriatic Sea, Greece, Ireland, western and southern coasts of Great Britain, and certain parts of France, Italy, Spain, and Norway, compound events could potentially contribute 20–25% or even more of all economic losses from flooding. In all aforementioned countries there are known examples of damaging floods contained in the HANZE database.

**Figure 7. Potential expected annual economic damage of coastal floods as % of GDP, 1950–2020, in constant 2020 exposure, per NUTS3 region. Potential impacts per region include all events above 100 ha flooded area threshold per NUTS3 region, including those not passing the socioeconomic impact thresholds.**





**Figure 8. Potential expected annual economic damage of riverine floods as % of GDP, 1950–2020, in constant 2020 exposure, per NUTS3 region. Potential impacts per region include all events above 100 ha flooded area threshold per NUTS3 region, including those not passing the socioeconomic impact thresholds.**





**Figure 9. Share of compound floods is total potential economic losses, 1950–2020, in constant 2020 exposure, per NUTS3 region. Potential impacts per region include all events above 100 ha flooded area threshold per NUTS3 region, including those not passing the socioeconomic impact thresholds. Individual riverine and coastal events contributing to compound events were excluded to compute this metric.**



**3.3 Validation**

**3.3.1 Extreme river discharges**

At least one river discharge station with adequate data length was available for 7742 events (63% of the total), and nearly 292,000 timeseries were identified within the NUTS3 regions potentially affected by those events. Most of the data is available for events that have occurred in the United Kingdom, Poland, Spain, Sweden, Germany, France, and Norway. The $R^2$ between modelled and observed peak discharge for all event time series, standardised by reported upstream area, is 0.45. However, the relative discharges are more of interest of this study, and modelled peak discharges corrected for difference in average annual discharges have an $R^2$ of 0.63. The timeseries of daily discharge during the events is good (0.5-0.75) or very good (0.75-1) for 59% of all station-events in terms of Spearman's $R^2$, and for 30% in terms of KGE score. On the other hand, poor (0-0.2) or very poor (<0) performance was recorded for 18% and 41% of stations, respectively. There is relatively little difference in performance depending on classification of events, except for far worse results for events classified as false positives ("E"). Here, the poor or very poor score was recorded for 83% of station-events, compared to 37% for HANZE flood events ("A"). Performance also varies strongly by location (Fig. 10), with e.g. Germany, Ireland, Austria, Belgium, and Slovakia recording much higher shares of good or very good station performance (above 40%) than e.g. Poland, Spain, Sweden, and Portugal (less than 25%).



**a**

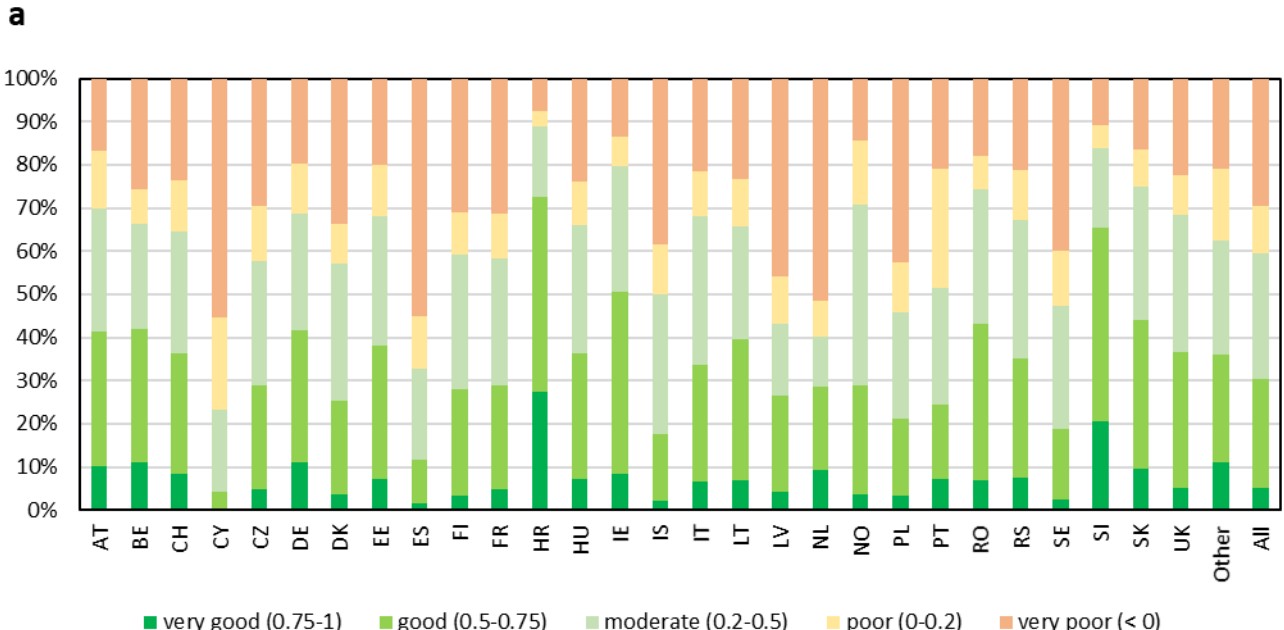

**b**

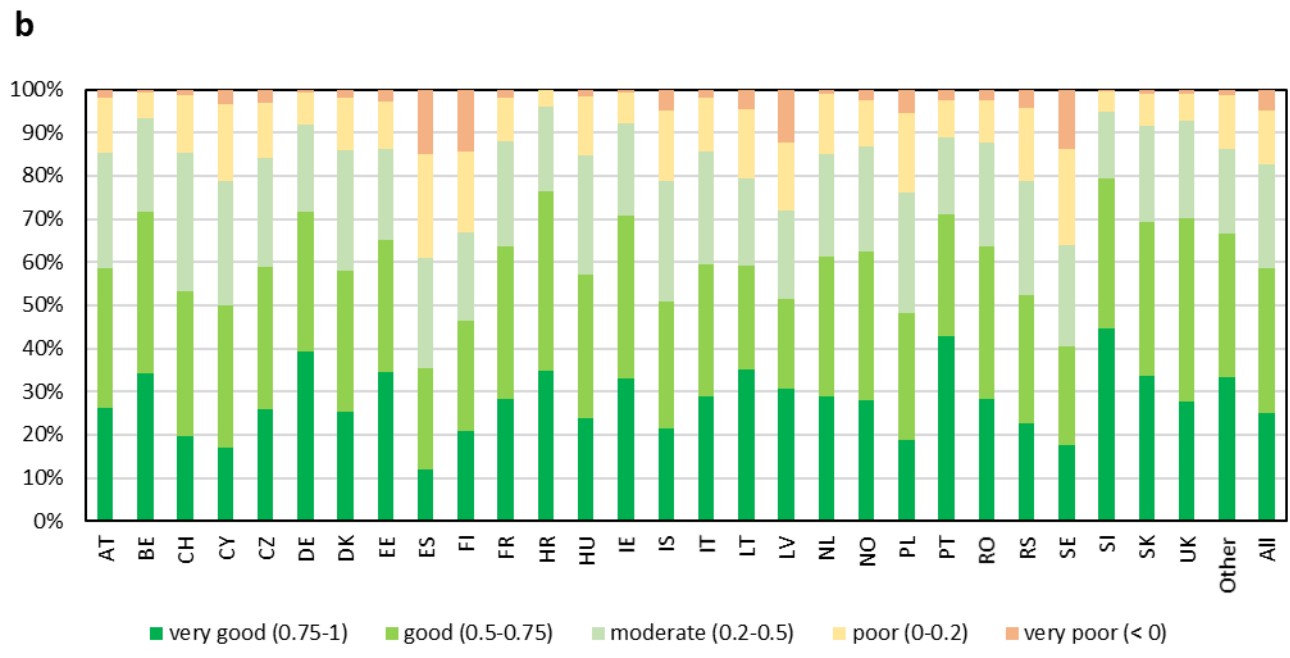

**Figure 10. Comparison of daily river discharge during flood events in the catalogue, or a 30-day window centred around the dates of the event. Abbreviations are NUTS level 0 country codes. The graph shows the percentage of all stations per country by performance class: (a) KGE score; (b) Spearman's coefficient of determination.**




**3.3.2 Extreme sea levels**

At least one tide gauge with adequate data length was available for 1363 events (56% of the total), and a total of 8102 time series were identified within the NUTS3 regions potentially affected by those events. Most of the data is available for events that have occurred in the United Kingdom, Denmark, Norway, the Netherlands, France, Sweden, and Germany. The overall results are compared using several metrics in Table 7. Overall, the maximum sea levels observed during the various potential coastal floods were well reproduced, with the main source of inaccuracies being storm surge heights. Further, 80% of modelled time series spanning the duration of the events indicated a good or very good $R^2$ when compared with observations. For tides and total water level, such performance was measured for 93–94% of stations. The best performance of the storm surge model was recorded for North and Baltic seas (Fig. 11), with far lower performance for the Eastern Mediterranean Sea. However, potential flood events and observational data are both relatively scarce in the latter region, which had the lowest scores also for reproducing tides and combined sea level. As in the case of riverine events, there is little variation between events by classification, though historical HANZE events ("A") had slightly higher scores for storm surge heights and combined sea level than all other classes. This could be, to some extent, the result of the difference in the geographical distribution of events.

**Table 7. Comparison between maximum hourly sea level and its components during flood events in the catalogue, or a 7-day window centred around the dates of the event.**

| Metric | Storm surge height | Tide elevation | Combined sea level |
|---|---|---|---|
| Pearson's $R^2$ | 0.75 | 0.99 | 0.96 |
| Spearman's $R^2$ | 0.74 | 0.95 | 0.94 |
| Nash-Sutcliffe Efficiency | 0.47 | 0.99 | 0.96 |
| Root mean squared error (RMSE) in metres | 0.30 | 0.14 | 0.26 |
| RMSE to standard deviation ratio | 0.53 | 0.11 | 0.21 |





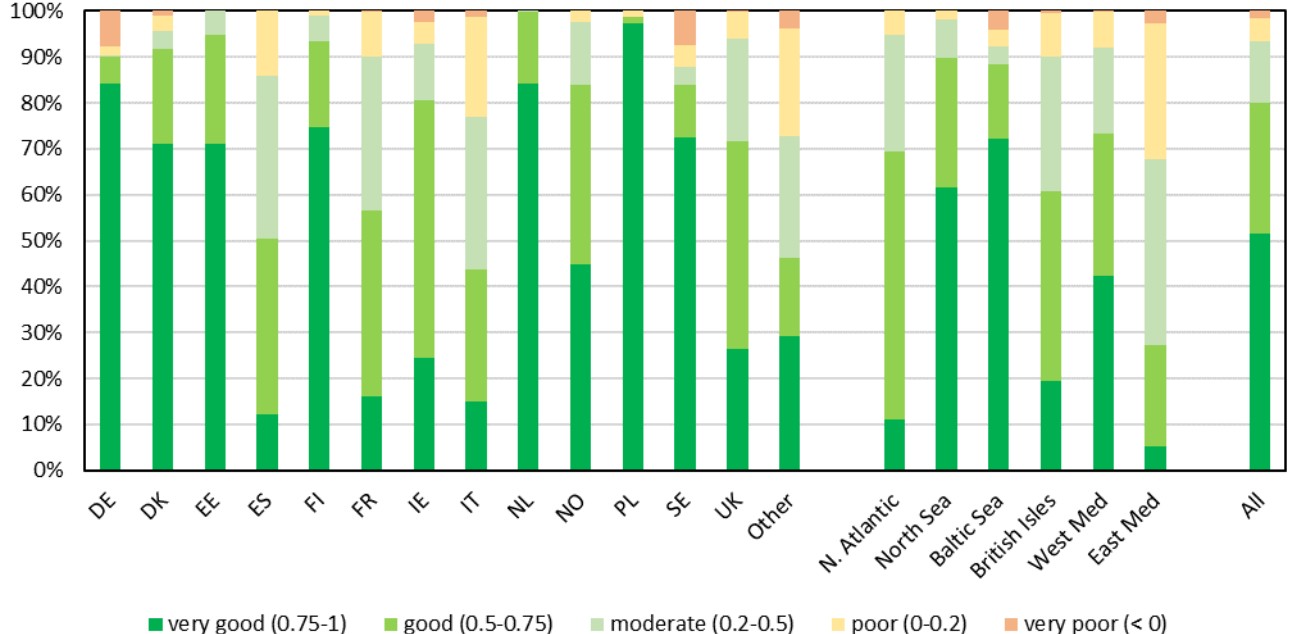

**Figure 11. Comparison between maximum hourly storm surge height during flood events in the catalogue, or a 7-day window centred around the dates of the event. The graph shows the percentage of all stations per country by performance of Pearson's $R^2$. Abbreviations in the left side of the graph are NUTS level 0 country codes. On the right side of the graph, stations are grouped by main European sea regions: "N. Atlantic" – exposed North Atlantic Ocean coasts (mostly France and Spain), "North Sea" – including Norwegian coasts, "Baltic Sea" – including Danish Straits, "British Isles" – coasts of Great Britain and Ireland, "West Med" and "East Med" – Western and Eastern Mediterranean Sea, respectively.**

### 3.3.3 Flood footprints

Comparison of modelled potential flood impacts with impacts based on satellite-derived flood footprints and actual impacts recorded in the HANZE database highlights the challenge of correctly recreating past floods (Table 8). For exactly half of the 20 floods for which a satellite-derived footprint is available, our modelled population affected were closer to reported population affected than estimates based on satellite-derived flood footprints, and vice versa. In most cases, satellite-derived footprints severely underestimated the extent of the flooding, with the exception of floods in the United Kingdom, where they indicated many times more affected population than the reported actual impact. In all cases the modelled area and persons affected were higher than the actual impact, as was the intention of the catalogue, as modelled without flood protection. However, there is a very close match in persons affected during the August 2002 flood in Czechia and Germany. In the whole catalogue, the area affected was higher than reported in 83% of cases where the actual impact was reported in HANZE (i.e. 256 out of 307), fatalities in 98% of cases (1473 out of 1496), population affected in 89% of cases (686 out of 773) and economic loss in 89% of cases (675 out of 755).





**Table 8. Comparison of modelled potential flood zone with satellite-derived footprints from the Global Flood Database (Tellman et**
**al., 2021) and reported impacts from HANZE (Paprotny et al., 2023) for several European floods, 2002–2015. Area flooded in km².**
**\* percentage of the satellite flood footprint reproduced by the modelled flood footprint of this study.**

| Event (country, month, year) | HANZE ID | Reported impacts (HANZE) | | Modelled impacts with potential flood zone | | Modelled impacts with satellite footprints | | Hit rate modelled area to satellite area* | Ratio of affected population | |
|---|---|---|---|---|---|---|---|---|---|---|
| | | Area flooded | Persons affected | Area flooded | Persons affected | Area flooded | Persons affected | | Modelled: reported | Reported: satellite |
| Albania, November/ December 2010 | 2031 | 139 | 24,700 | 894 | 91,776 | 194 | 8,260 | 56% | 3.7 | 3.0 |
| Austria, March 2006 | 21 | | 1,840 | 263 | 15,130 | 68 | 1,659 | 45% | 8.2 | 1.1 |
| Bosnia and Herzegovina, April-May 2004 | 2053 | 200 | 20,000 | 734 | 147,114 | 75 | 1,023 | 44% | 7.4 | 19.6 |
| Czechia, August 2002 | 86 | | 225,000 | 1247 | 225,513 | 90 | 4,018 | 54% | 1.0 | 56.0 |
| France, September 2002 | 244 | | 12,000 | 763 | 116,813 | 95 | 1,595 | 30% | 9.7 | 7.5 |
| France, December 2003 | 250 | | 27,000 | 1,843 | 245,870 | 767 | 11,954 | 67% | 9.1 | 2.3 |
| Germany, August 2002 | 341 | | 330,000 | 3,371 | 372,649 | 681 | 10,081 | 74% | 1.1 | 32.7 |
| Greece, January/ February 2015 | 403 | 250 | 500 | 405 | 3,696 | 268 | 256 | 44% | 7.4 | 2.0 |
| Hungary, March-May 2006 | 421 | 2,440 | 5,400 | 5,201 | 310,750 | 918 | 10,886 | 37% | 57.5 | 0.5 |
| Hungary, May/June 2010 | 422 | 1,230 | 5,000 | 1,376 | 77,306 | 199 | 214 | 85% | 15.5 | 23.3 |
| Italy, November/ December 2002 | 952 | | 10,000 | 2,031 | 424,594 | 119 | 29,321 | 13% | 42.5 | 2.9 |
| Italy, January 2003 | 954 | | 40,000 | 370 | 43,917 | 35 | 392 | 18% | 1.1 | 102.0 |
| Lithuania, March/April 2010 | 2200 | 400 | 2,000 | 1,211 | 27,851 | 214 | 464 | 59% | 13.9 | 4.3 |
| Montenegro, December 2010 | 2209 | | 6,630 | 289 | 21,390 | 198 | 2,330 | 34% | 3.2 | 2.8 |
| Poland, May/June 2010 | 1065 | 5,540 | 280,000 | 7,151 | 775,536 | 348 | 9,757 | 71% | 2.8 | 28.7 |
| Romania, July 2005 | 1148 | 993 | 58,700 | 1,664 | 85,918 | 338 | 1,061 | 50% | 1.5 | 55.3 |
| Romania, April/May 2006 | 1153 | 1,165 | 15,011 | 5,305 | 115,330 | 3,415 | 6,626 | 43% | 7.7 | 2.3 |
| UK, November/ December 2012 | 1558 | | 4,400 | 1,156 | 132,320 | 869 | 265,903 | 12% | 30.1 | 0.02 |
| UK, December 2013- February 2014 | 1561 | 450 | 25,000 | 828 | 225,781 | 815 | 388,930 | 13% | 9.0 | 0.06 |
| UK, December 2015- January 2016 | 1563 | | 64,000 | 1,016 | 100,633 | 1,472 | 480,026 | 9% | 1.6 | 0.13 |


Direct comparison between modelled and satellite footprints (Fig. 12) has shown that the hit rate, i.e. share of the satellite
footprints correctly reproduced by the model, varied between 30 and 85%, except for events in Italy and the United Kingdom,



where it was only 9–18%. However, the satellite footprints also performed very poorly against reported losses for those floods.
Some additional flood events were analysed, but were not included in Table 6 as the satellite footprints showed virtually no
population affected, which is in large contrast to actual impacts. Such a situation occurred e.g. for the summer floods in the
United Kingdom in 2007 that flooded homes of about 192,000 people (HANZE database number 1546), almost none of which
could be reproduced with satellite flood footprints.




**Figure 12. Example comparison between modelled and satellite-derived flood footprint of the 2006 event in Romania.**
**4 Discussion**
**4.1 Uncertainties and limitations of the models and modelled data**
The elaborate modelling chain involving both riverine and coastal processes is subject to multiple cascading limitations and
uncertainties. The starting point of the simulations are input climate data, derived from global reanalyses. Though ERA5 and



ERA5-Land are state-of-the-art, they still encounter problems of inhomogeneities, gaps or errors in observational data, model
biases, and limitations in representing precipitation extremes in particular (Hersbach et al., 2020, Muñoz-Sabater et al., 2021).
In the case of the riverine model, bias-adjustment and downscaling was carried out, but it is also only a statistical transformation
that depends on the quality of high-resolution observations as well (see section 4.1.2).
Validation results in section 3.1 indicate mostly good performance of the models in reconstructing past extreme discharges
and sea levels, but not in all areas. Some regions are more challenging to model than others, e.g. due to complex topography
or shoreline, or strong anthropogenic influence on the water cycle (especially through reservoirs). Not all types of floods or
processes that drive them could be represented. Most noticeably, the resolution of the riverine model is inadequate to capture
smaller flash floods, as the hydrological model has a spatial resolution of 1' driven by climate data that was twice downscaled
(first from ERA5 to ERA5-Land, then using ISIMIP3BASD method) and with a temporal resolution of six hours. Additionally,
flood hazard maps used to generate the footprints only covered catchments with an upstream area of at least 100 km$^2$.
Consequently, 91% of slow-onset riverine floods from HANZE were reproduced, but only 55% of flash floods. Urban floods
are not represented at all (also in the HANZE dataset).
Further, no flood defences are represented in the model, which is by design, as information on this aspect is scarce, especially
in the temporal dimension. At the same time, a flood that was historically prevented by existing defences might not have been
prevented under counterfactual conditions. We also hypothesise that flood protection levels are driven to some extent by flood
risk and flood occurrence (section 4.2). The use of flood hazard maps for a defined set of scenarios enables generating flood
footprints without carrying out a computationally infeasible continuous hydrodynamic simulation over a period of 71 years.
However, the maps assume a specific hydrograph which is not necessarily valid for all floods with the same peak discharge.
Further, the three sets of maps (including two sets for different catchment sizes) are methodologically different and were
created for diverse sets of scenarios. Whereas the coastal maps were rerun specifically for this study based on the results of
the extreme sea level modelling, the riverine maps are from previous studies. Their application is in some locations problematic
due to inconsistencies in river network delineation between EFAS and the hazard maps. The accuracy of the riverine flood
hazard maps is also variable depending on the region and the probability of occurrence (see Paprotny et al., 2017, and Dottori
et al., 2022, for details).
Compound floods are represented by merging riverine and coastal flood zones, which neglects the possible interaction between
the storm surge and river discharge that could generate higher water levels than is possible for individual drivers. Additionally,
not all coastal processes are included in the model, such as interaction between tide and storm surges, or influence of SLR on
tide elevations. Wave run-up is only approximated by taking one-fifth of offshore significant wave height, as more precise
estimates would require a very detailed model of the nearshore. Finally, long-term land motion is limited to GIA due to lack
of detailed data on the subject.





## 4.2 Uncertainties and limitations of the observations and documentary sources

The results are influenced not only by the accuracy of models, but also that of the observations. Our river discharge simulations are driven by reanalysis data that were downscaled and bias-adjusted using interpolated meteorological observations, the accuracy of which depends strongly on the density of point meteorological data. As shown in Thiemig et al. (2022), precipitation during extreme events in the EMO dataset can at times diverge strongly from other reported measurements. Though our meteorological input data is still driven primarily by ERA5, the reanalysis itself is influenced by availability of meteorological data, which is very inhomogeneous in time (Hersbach et al., 2020). This might be the reason for the noticeably lower performance of our model in reproducing flood events in the 1950s.

Model calibration and validation, as well as classification of the flood event catalogue is affected by the availability of tide and river gauges (section 3.1.1 and 3.1.2). The data is unevenly distributed, with most data available for northern Europe, particularly the Nordic countries and the British Isles. On the other hand, very limited data was available for Italy, Greece, and Balkan countries. It is further uneven in time, with both the 1950s and the last few years until 2020 having lower coverage than the 1990s and 2000s in particular. Identification of events as false positives ("E") is also potentially problematic, as in large NUTS3 regions the only available observations could be outside the impact zone of the event, hence incorrectly suggesting that the model generated a 'bogus' event. Satellite-derived footprints were used to validate the modelled flood footprints, but themselves often widely diverged from reported impacts. The hit rate between satellite and model data varied significantly between individual events, similarly observed in a reconstruction of recent European coastal floods by Le Gal et al. (2023).

Similarly, documentary sources on socioeconomic impacts of floods are highly uneven in quality between countries. For instance, while there are comprehensive databases and flood catalogues accessible e.g. for France, Italy, Norway, Portugal, Spain, or Switzerland, and even some Balkan countries, scattering of information makes it very laborious to collect data for other countries, e.g. Austria, Germany, and the United Kingdom. Many compilations of flood impacts only cover the recent two decades, while older flood catalogues published in the 1980s or 1990s often have no newer follow-ups. This strongly affects the frequency of "C" (No impacts) events relative to "D" (Impacts unknown). Thanks to extensive research in the HANZE database, this has less effect on detection of "A" (Impacts, data) and "B" (Impacts, no data) events. Still, uncertainty surrounds designation of flood events as having "significant" socioeconomic impacts. The thresholds defined in HANZE are somewhat arbitrary, though based on experience of collecting more than 2500 records in the dataset. In case of smaller events, their classification is uncertain if the data is incomplete or not very accurate. This is potentially problematic for "B" events, where at times no quantitative data at all is available, and the classification was based on the description of impacts only. Finally, NUTS3 regions, the principal socioeconomic unit of observation here and in HANZE, vary in size both in terms of area and population. It might be slightly easier for floods in large regions to pass region-scale threshold for minimum flood area in the model, and to be considered affected in HANZE, where region-scale impact thresholds are also applied when detailed damage data are available.



**5 Conclusions**

This study is the largest attempt to reconstruct past flood losses in Europe, and makes an advance towards full decomposition of drivers of historical flood losses. We created a flood catalogue for Europe containing 14,699 events with significant socioeconomic impact potential. It covers riverine, coastal and compound events over a period of 71 years, and considers climate change, evolving human impact on catchments, and growing exposure. The catalogue includes 1504 out of 2037 damaging floods since 1950 included in HANZE dataset (Paprotny et al., 2023), including some 90% of coastal, compound and slow-onset riverine floods, and 55% of flash floods. The coverage of reported impacts of those events is 81-99% depending on the exact measure. The performance of the model is relatively stable over time, though slightly worse for the 1950s.

The flood catalogue was primarily devised as the baseline (factual) reconstruction of past floods in Europe. However, it can be also used directly for multiple applications. The immediate follow-up to this analysis will be modelling changes in flood preparedness in Europe in the past 70 years, including flood protection standards and relative losses (vulnerability). The modelling chain can be further used with counterfactual climate inputs. Methods such as ATTRICI (Mengel et al., 2021) enable removing the global warming effect from all variables required to model riverine discharges. Additional counterfactual simulations are possible to quantify the human influence on catchments, particularly through construction of reservoirs (Boulange et al., 2021). Methods such as transformed-stationary extreme value analysis (Mentaschi et al., 2016) can be used to detrend storm surge heights in addition to removing the long-term sea level rise. Together with HANZE historical exposure maps (Paprotny and Mengel, 2023), counterfactual scenarios for all components of risk would be achieved. This would provide the first comprehensive impact attribution of European flood losses and generate an important reference dataset for pan-European flood risk assessments.

*Data & code availability:* Numerous public datasets and models were used in the study, results of which are also publicly available. Details were to find each dataset and model are provided in Appendix A3.

*Author contributions.* DP developed the concept, implemented the methods, collected and processed most of the data, and acquired funding. BR collected part of the historical impact data and performed part of the flood event classification. MV computed coastal flood hazard maps. PT and JS performed the validation based on satellite-derived flood footprints and created the online visualisation of the study. FD and ST contributed datasets and methods for the riverine and coastal simulations, respectively. LF and HK helped to develop the concept and methods. All authors wrote the paper.

*Competing interests.* The authors declare that they have no conflict of interest.

*Financial support.* This research has been supported by the German Research Foundation (DFG) through project "Decomposition of flood losses by environmental and economic drivers" (FloodDrivers), grant no. 449175973.



**Appendix A1. Contents of "B" list of historical floods**

The format of the database of "B" list events follows the same format of HANZE database (Paprotny et al., 2023), with a reduced number of fields as events were confined to the "B" list specifically due to lack of relevant data (primarily flood impact statistics). Most fields have strictly-defined permitted values, except "Notes", which includes explanation why impacts should be considered significant (using partial available data or descriptive indicators), and "Data sources" which lists all cited references. The latter are often the same as used in the HANZE database, therefore only publications specific to the B list are included in the full bibliographic details provided with the event file. For detailed discussion about the contents of each field we refer to Paprotny et al. (2023).

**Table A1. Summary of fields recorded in the "B" list of floods.**

| Variable | Short description | Field type | Permitted values |
|---|---|---|---|
| ID | Unique event identifier | integer | 7000…8999 |
| Country code | Two-letter country code | string | Codes from Table B1 |
| Year | Year of the event | integer | 1950…2020 |
| Country name | Country name | string | Names from Table B1 |
| Start date | Daily start date | date | 1.1.1950…31.12.2020 |
| End date | Daily end date | date | 1.1.1950…31.01.2021 |
| Type | Detailed type of event | string | River, Flash, Coastal, River/Coastal |
| Regions affected (NUTS3 v2010) | Regions were human or economic losses were reported, at NUTS3 level (version 2010) | string | Codes from Table B2 |
| Regions affected (NUTS3 v2021) | As above, but at using NUTS version 2021 | string | Codes from Table B3 |
| Notes | Other relevant information or notes on issues with the data | string | Free text |
| References | List of publications and databases from which the information was obtained | string | Free text |



**Appendix A2. Contents of the modelled flood event catalogue**

**Table A2. Summary of fields recorded in the modelled flood event catalogue.**

| Variable | Short description |
|---|---|
| ID | Unique event identifier |
| Country code | Two-letter country code |
| Year | Year of the event |
| Country name | Country name |
| Start date | Daily start date |
| End date | Daily end date |
| Type | Detailed type of event |
| Flood source | Rivers or sea basins in the potentially-affected area (from Vogt et al., 2007, and Fourcy and Lorvelec, 2013) |
| Regions affected (NUTS3 v2010) | Regions were human or economic losses were reported, at NUTS3 level (version 2010) |
| Regions affected (NUTS3 v2021) | As above, but at using NUTS version 2021 |
| Area inundated | Potential inundated area in hectares (ha) |
| Fatalities, YE | Potential fatalities, in persons, year-of-event exposure |
| Fatalities, 1950 | Potential fatalities, in persons, 1950 exposure |
| Fatalities, 2020 | Potential fatalities, in persons, 2020 exposure |
| Persons affected, YE | Potential persons affected, in persons, year-of-event exposure |
| Persons affected, 1950 | Potential persons affected, in persons, 1950 exposure |
| Persons affected, 2020 | Potential persons affected, in persons, 2020 exposure |
| Economic loss, YE | Potential direct economic loss, in thousands of 2020 euros, year-of-event exposure |
| Economic loss, 1950 | Potential direct economic loss, in thousands of 2020 euros, 1950 exposure |
| Economic loss, 2020 | Potential direct economic loss, in thousands of 2020 euros, 2020 exposure |
| Loss threshold | Threshold for direct economic losses applied to the event, in thousands of 2020 euros |
| Mean water depth | Average water depth in the potential inundated zone |
| Return period | Average (geometric) of return periods along potential affected river grid cells or coastal segments, from detrended 1950–2020 data, Generalised Pareto distribution |
| Hydro data | Indicates if river or tide gauge data were available for this event (1 – yes, 0 – no) |
| RP2 exceedance | Indicates if a 2-year return period was exceeded in the observational data (1 – yes, 0 – no) |
| Category | Classification of event according to Table 4 |
| HANZE ID | Flood event ID if event classified as "A" or "B", otherwise empty field |



**Table A3. Availability of input and output data and models from the study. Models are indicated by *italics*.**

| Variable, data | Dataset/*model* | Resource link |
|---|---|---|
| River discharges | HERA | https://data.jrc.ec.europa.eu/dataset/a605a675-9444-4017-8b34-d66be5b18c95 |
| Meteorological data for storm surge simulation, significant wave height | ERA5 | https://doi.org/10.24381/cds.e2161bac |
| *Hydrodynamic model (coastal)* | *Delft3D* | https://oss.deltares.nl/web/delft3d/get-started |
| Tide elevation constituents | FES2014 | https://www.aviso.altimetry.fr/en/data/products/auxiliary-products/global-tide-fes.html |
| *Tide elevation model* | *pyTMD* | https://github.com/tsutterley/pyTMD |
| Mean dynamic topography | Global Ocean Mean Dynamic Topography | https://doi.org/10.48670/moi-00150 |
| Sea level rise | Hourly Coastal water levels with Counterfactual | https://zenodo.org/records/7771386 |
| Sea level rise | European Seas Gridded L 4 Sea Surface Heights | https://doi.org/10.48670/moi-00141 |
| Sea level rise | Global Ocean Gridded L 4 Sea Surface Heights | https://doi.org/10.48670/moi-00148 |
| Glacial isostatic adjustment | ICE-6G_C | https://www.atmosp.physics.utoronto.ca/~peltier/data.php |
| Storm surge heights, combined water level, tide levels | This study | https://doi.org/10.5281/zenodo.10630338 |
| DEM for coastal inundation | GLO-30 | https://doi.org/10.5069/G9028PQB |
| *Hydrodynamic model for coastal inundation* | *Lisflood-ACC* | *https://www.seamlesswave.com/LISFLOOD8.0* |
| Land use and population at 100 m resolution | HANZE v2.0 output maps | https://doi.org/10.5281/zenodo.7885990 |
| *Exposure model (population, fixed assets by sector)* | *HANZE v2.0* | https://doi.org/10.5281/zenodo.7556953 |
| Historical flood impacts ("A" list) and list of references | HANZE v2.1 | https://doi.org/10.5281/zenodo.8410025 |
| Significant flood events without direct impact data ("B" list) | This study | https://doi.org/10.5281/zenodo.10629443 |
| List of documentary sources used | This study | https://doi.org/10.5281/zenodo.10629443 |
| Coastal flood hazard maps, flood catalogue input data | This study | https://doi.org/10.5281/zenodo.10630862 |
| River flood hazard maps | JRC maps | https://doi.org/10.2905/1D128B6C-A4EE-4858-9E34-6210707F3C81 |
| River flood hazard maps | RAIN project maps | https://doi.org/10.4121/uuid:968098ce-afe1-4b21-a509-dedaf9bf4bd5 |
| Historical flood database | EM-DAT | https://public.emdat.be/ |
| Historical flood database | EEA Flood Phenomena | https://www.eea.europa.eu/data-and-maps/data/european-past-floods/flood-phenomena |
| Historical flood database | Dartmouth Flood Observatory | http://floodobservatory.colorado.edu/Archives/index.html |
| Historical flood database | FFEM-DB | https://doi.org/10.4121/14754999.v2 |
| Historical flood database | Recorded Flood Outlines | https://www.data.gov.uk/dataset/16e32c53-35a6-4d54-a111-ca09031eaaaf/recorded-flood-outlines |
| River discharge data | GRDC | https://portal.grdc.bafg.de/ |
| River discharge data (France) | HydroPortail | https://www.hydro.eaufrance.fr/rechercher/entites-hydrometriques |





| River discharge data (Norway) | NVE, Historiske vannføringsdata til produksjonsplanlegging | https://www.nve.no/vann-og-vassdrag/hydrologiske-data/historiske-data/historiske-vannfoeringsdata-til-produksjonsplanlegging/ |
|---|---|---|
| River discharge data (Spain) | Centro de Estudios Hidrográficos, Anuario de aforos | https://ceh.cedex.es/anuarioaforos/default.asp |
| River discharge data (Sweden) | SMHI Vattenweb | https://www.smhi.se/data/hydrologi/vattenwebb |
| River discharge data (UK) | UK National River Flow Archive | https://nrfa.ceh.ac.uk/ |
| River discharge, sea level data (Poland) | IMGW-PIB, Dane Publiczne | https://danepubliczne.imgw.pl/ |
| Sea level data | GESLA v3 | https://gesla787883612.wordpress.com/ |
| Sea level data | Poseidon System | https://poseidon.hcmr.gr/services/ocean-data/situ-data |
| Satellite flood footprints | Global Flood Database | https://global-flood-database.cloudtostreet.ai/#interactive-map |
| *Flood catalogue generation model* | *This study* | https://doi.org/10.5281/zenodo.10678820 |
| Modelled flood catalogue | This study | https://doi.org/10.5281/zenodo.10629443 |
| Modelled flood footprints | This study | https://doi.org/10.5281/zenodo.10640692 |

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
