# Peer review of "Merging modelled and reported flood impacts in Europe in a"

_EGUsphere, 2024_

## Referee Comment (RC1)

**Technical Comments**

Line 123: For clarity it would be helpful to include the end date of the model run (e.g., the model was run from 1 January 1949 to XXXX, with the first year...)

Line 140, Table 2: I suggest adding the temporal resolution of the data as a column of the data in this table (when applicable). Edit width for spatial resolution column.

Line 189: Same comment as line 123, add full time period of simulation.

Line 236: To clarify, what was the total number of events modeled for each type of event? This modeling effort is impressive, and the number of total events modeled (prior to filtering based on impacts) would be helpful to highlight more clearly in the methods but also in the introduction or even abstract.

Line 302, Table 4: Either in the text or as a column in the table, it would be helpful to explicitly state which classes are included in the final catalogue. If all classes are included in the catalogue that would also be helpful to state in the text. Edit width for class column.

Line 314: How do these thresholds compare with the thresholds mentioned in Table 3?

Line 386: Text has values 11.7%, 5.4% and 3.7% for each event but it might also be helpful here to give the total number of events by event type. I would suggest including the total number of events modeled by type and then the total number of events included in the final catalogue by event type. These numbers are present throughout the text but highlighting them more explicitly (whether in this section or in introduction) would help demonstrate the scale of modeling efforts completed for this paper.

Line 388: The values referenced in this line are the addition of pie chart slices in Figure 2. It might be helpful to create a 4$^{th}$ pie chart that has the classification breakdown by all event types.

Table 8: I would suggest changing the ratio of affected population in the 'Reported: Satellite' column to 'Satellite: Reported' to be comparable with the 'Modeled: Reported' column. Edit width of HANZE ID column in table.

---

## Author Comment (AC1)

Below, we respond (**R**) to the more detailed comments (**C**) of the reviewer.

**Comments/Questions**

**C1:** To improve the clarity of the steps included in the methodology section of the paper, I would suggest converting Table 1 to a flow diagram. Examples of such figures are included in Bates et al. 2021 (Figure 1) and Collins et al. 2022 (Figure 1). This modification would provide a visual and concise overview of the models, data, and filtering used within the different stages of the method section.

**R1:** In the revision, we will construct a diagram based on the papers suggested, as it would indeed improve the presentation of our methodology.

**C2:** In reading through the methods section of the paper, I had a question in section '2.2.4 Deriving coastal flood footprints' regarding the use of return periods for modeled depths and extents of identified flood events. In this section the text mentions that return periods (2, 5, 10, 20, 30, 50, 100, 200, and 500 years) are used for coastal inundation modeling at each coastal segment using Lisflood-ACC at 30m resolution spanning 200km landwards. Then in Line 162 the text states "Total water level of each segment-level flood event is linked with the water level used to generate flood hazard maps for each segment."

Hypothetically, does this mean that for a coastal segment with an event where the total water level is 15 ft, the depths of water for the flooded area of this event are interpolated between return periods? For example, if the 10-year return period has a water level of 10ft and the 20-year return period has a water level of 20 ft; then the depths associated with an event with a water level of 15 ft at that segment would be the mean depth between the 10-year and 20-year return period maps? Furthermore, are the extents of these hazard maps consistent between return periods? If not, how is the area of inundation interpolated between return periods? These questions aim to clarify how flooded area and depths are interpolated between return periods. I have similar clarification questions regarding interpolation between return period hazard maps for section '2.3.4 Deriving riverine and compound flood footprints.'

**R2:** In general, the interpolation works as described by the reviewer: water depth is interpolated based on water level of the event and scenarios used to derive the 'lower' and 'upper' hazard map. One thing we forgot to mention in 2.3.4 is that, due to the logarithmic nature of the relationship between river discharge and water level, we used the natural logarithm of discharge as basis of interpolation. The maps different extents, therefore if an area is not flooded in the 'lower' map, the interpolation is between zero depth and water depth of the 'upper' map is made. This might slightly overstate the extent in the interpolated footprint, however the effect should be small as anyway we only consider water depth of at least 0.1 m as flooded area in further processing (as in the original riverine flood maps). We will clarify those details in the revision.

**C3:** In the results section '3.2.1 Temporal changes in potential flood impacts' there are observed increases in both the number and impact of events across all three event types shown in Table 6. However, the text in this section references percent changes and values that are not present in Table 6. To enhance clarity of results, it would be helpful to reference the values included in Table 6. For example, in Lines 469-270, based on the information provided in Table 6, the sentence should read as follows: "…they are equivalent to at least a 164% increase in potential coastal flood losses in an average year between 1950 and 2020 in the case of fatalities, 852% in the case of economic loss, and 83% in the case of affected population." If the current figures in the text are accurate, clarification on how these values were calculated would be valuable to improve clarity of the

magnitude of these trends. Additionally, according to Table 6, the potential impacts for compound events appear to have increased more substantially than riverine and coastal events while the opposite is indicated in Lines 471-472.

**R3:** Indeed, there is an inconsistency in presenting the data between Table 6 and the text. However, it is because the table uses annual rate of increase, while in the text we converted it into cumulative increase over 1950-2020, i.e. 75% increase in population affected by coastal floods is equivalent to 0.8% annual increase in Table 6. The first paragraph refers to "Annual increase of potential impacts (%)" in the table, and not "Increase in total impacts relative to 1950 exposure". The former indicates the increase in losses under different exposure scenarios, while the latter indicates only the effect of exposure growth. Therefore, the text should have read: "…they are still equivalent to at least 0.3% annual increase in potential coastal flood losses between 1950 and 2020 in case of fatalities, 0.5% in case of economic loss and 0.8% in case of affected population." In this context, compound flood risk increases more than coastal or riverine, which should be described as "For riverine floods, the potential impacts have grown even more, while the strongest increase is indicated for compound floods, at a rate of at least 1.9% per year since 1950." In the revision, we will modify the text to bring it fully in line with Table 6.

**Technical Comments**

**C4:** Line 123: For clarity it would be helpful to include the end date of the model run (e.g., the model was run from 1 January 1949 to XXXX, with the first year…)

**R4:** We will clarify in the text that the model was run until 31 December 2020.

**C5:** Line 140, Table 2: I suggest adding the temporal resolution of the data as a column of the data in this table (when applicable). Edit width for spatial resolution column.

**R5:** We will modify the table to clarify the temporal resolution as follows:

| Component | Source | Temporal resolution |
|---|---|---|
| Storm surge height | Delft3D simulation (this study) | hourly |
| Tide elevation | FES2014 | hourly |
| Wave run-up | ERA5 | hourly |
| Mean dynamic topography | Global Ocean Mean Dynamic Topography | 1993-2012 average |
| Sea level rise | 1950–99: Hourly Coastal water levels with Counterfactual (HCC) | hourly (used as annual average) |
| | 2000–2020: European Seas Gridded L4 Sea Surface Heights* | monthly (used as annual average) |
| | 2000–2020: Global Ocean Gridded L4 Sea Surface Heights* | |
| Glacial isostatic adjustment | ICE-6G_C | long-term rate of change |

**C6:** Line 189: Same comment as line 123, add full time period of simulation.

**R6:** We will clarify in the text that the model was run until 31 December 2020.

**C7:** Line 236: To clarify, what was the total number of events modeled for each type of event? This modeling effort is impressive, and the number of total events modeled (prior to filtering based on impacts) would be helpful to highlight more clearly in the methods but also in the introduction or even abstract.

**R7:** We will add the information on the number of events by type at the different stages, as shown in the table below:

| Event type | Coastal | Riverine | Compound | Total |
|---|---|---|---|---|
| NUTS3-level events | 22,446 | 213,517 | 5235 | 241,198 |
| Spatiotemporarily aggregated events | 4208 | 19,918 | 1452 | 25,578 |
| Filtered events by impact | 2436 | 11,205 | 1058 | 14,699 |

**C8:** Line 302, Table 4: Either in the text or as a column in the table, it would be helpful to explicitly state which classes are included in the final catalogue. If all classes are included in the catalogue that would also be helpful to state in the text. Edit width for class column.

**R8:** We will clarify that all classes are included in the final catalogue, even false positives, so that users of the dataset can decide whether to use all data or limit it to certain classes.

**C9:** Line 314: How do these thresholds compare with the thresholds mentioned in Table 3?

**R9:** The thresholds were defined in previous research (HANZE v1, Paprotny et al. 2018) based on analysis of reported flood impact data to avoid inclusion of insignificant, highly localized events that would not be reproducible in pan-European flood models. The potential impact thresholds had to be much higher as the potential flood catalogue does not include flood protection, and was devised iteratively to maximize the coverage of historical events without creating too many non-impact events.

**C10:** Line 386: Text has values 11.7%, 5.4% and 3.7% for each event but it might also be helpful here to give the total number of events by event type. I would suggest including the total number of events modeled by type and then the total number of events included in the final catalogue by event type. These numbers are present throughout the text but highlighting them more explicitly (whether in this section or in introduction) would help demonstrate the scale of modeling efforts completed for this paper.

**R10:** We will highlight more clearly in the text the absolute number of events by type and class. Again, all classes are included in the published data of 15,000 events, which as highlighted in response to comment 7, was filtered from almost 26,000 events to include only those with significant potential to cause impacts.

**C11:** Line 388: The values referenced in this line are the addition of pie chart slices in Figure 2. It might be helpful to create a 4th pie chart that has the classification breakdown by all event types.

**R11:** We will add an additional graph to include all event types:

[Figure]

(d) all floods

Legend:
- A: Impacts, data
- B: Impacts, no data
- C: No impacts
- D: Unknown impacts
- E: False positive
- F: No information

Values shown: 1270, 207, 4041, 5798, 557, 2826

**C12:** Table 8: I would suggest changing the ratio of affected population in the 'Reported: Satellite' column to 'Satellite: Reported' to be comparable with the 'Modeled: Reported' column. Edit width of HANZE ID column in table.

**R12:** We will edit the table as suggested by the reviewer.

---

## Author Response (AR1)

We would like to thank the reviewers and the editor for taking the time to read our rather long paper and for the many important comments, which have enabled us to clarify important methodological points, and further for the overall positive assessment. Below, we respond (**R**) to the more detailed comments (**C**) of the reviewers and the editor.

**Reviewer #1**

**Comments/Questions**

**C1:** To improve the clarity of the steps included in the methodology section of the paper, I would suggest converting Table 1 to a flow diagram. Examples of such figures are included in Bates et al. 2021 (Figure 1) and Collins et al. 2022 (Figure 1). This modification would provide a visual and concise overview of the models, data, and filtering used within the different stages of the method section.

**R1:** We have prepared a diagram (Fig. 1) based on the papers suggested to replace Table 1.

**C2:** In reading through the methods section of the paper, I had a question in section '2.2.4 Deriving coastal flood footprints' regarding the use of return periods for modeled depths and extents of identified flood events. In this section the text mentions that return periods (2, 5, 10, 20, 30, 50, 100, 200, and 500 years) are used for coastal inundation modeling at each coastal segment using Lisflood-ACC at 30m resolution spanning 200km landwards. Then in Line 162 the text states "Total water level of each segment-level flood event is linked with the water level used to generate flood hazard maps for each segment."

Hypothetically, does this mean that for a coastal segment with an event where the total water level is 15 ft, the depths of water for the flooded area of this event are interpolated between return periods? For example, if the 10-year return period has a water level of 10ft and the 20-year return period has a water level of 20 ft; then the depths associated with an event with a water level of 15 ft at that segment would be the mean depth between the 10-year and 20-year return period maps? Furthermore, are the extents of these hazard maps consistent between return periods? If not, how is the area of inundation interpolated between return periods? These questions aim to clarify how flooded area and depths are interpolated between return periods. I have similar clarification questions regarding interpolation between return period hazard maps for section '2.3.4 Deriving riverine and compound flood footprints.'

**R2:** In general, the interpolation works as described by the reviewer: water depth is interpolated based on water level of the event and scenarios used to derive the 'lower' and 'upper' hazard map. One thing we forgot to mention in 2.3.4 is that, due to the logarithmic nature of the relationship between river discharge and water level, we used the natural logarithm of discharge as basis of interpolation. The maps have different extents, therefore if an area is not flooded in the 'lower' map, the interpolation is between zero depth and water depth of the 'upper' map is made. This might slightly overstate the extent in the interpolated footprint, however the effect should be small as anyway we only consider water depth of at least 0.1 m as flooded area in further processing (as in the original riverine flood maps). We now clarified this in the text.

**C3:** In the results section '3.2.1 Temporal changes in potential flood impacts' there are observed increases in both the number and impact of events across all three event types shown in Table 6. However, the text in this section references percent changes and values that are not present in

Table 6. To enhance clarity of results, it would be helpful to reference the values included in Table 6. For example, in Lines 469-270, based on the information provided in Table 6, the sentence should read as follows: "…they are equivalent to at least a 164% increase in potential coastal flood losses in an average year between 1950 and 2020 in the case of fatalities, 852% in the case of economic loss, and 83% in the case of affected population." If the current figures in the text are accurate, clarification on how these values were calculated would be valuable to improve clarity of the magnitude of these trends. Additionally, according to Table 6, the potential impacts for compound events appear to have increased more substantially than riverine and coastal events while the opposite is indicated in Lines 471-472.

**R3:** Indeed, there is an inconsistency in presenting the data between Table 6 and the text. However, it is because the table uses annual rate of increase, while in the text we converted it into cumulative increase over 1950-2020, i.e. 75% increase in population affected by coastal floods is equivalent to 0.8% annual increase in Table 6. The first paragraph refers to "Annual increase of potential impacts (%)" in the table, and not "Increase in total impacts relative to 1950 exposure". The former indicates the increase in losses under different exposure scenarios, while the latter indicates only the effect of exposure growth. Therefore, the text should have read: "…they are still equivalent to at least 0.3% annual increase in potential coastal flood losses between 1950 and 2020 in case of fatalities, 0.5% in case of economic loss and 0.8% in case of affected population." In this context, compound flood risk increases more than coastal or riverine, which should be described as "For riverine floods, the potential impacts have grown even more, while the strongest increase is indicated for compound floods, at a rate of at least 1.9% per year since 1950." We modified the text to bring it fully in line with Table 6.

**Technical Comments**

**C4:** Line 123: For clarity it would be helpful to include the end date of the model run (e.g., the model was run from 1 January 1949 to XXXX, with the first year…)

**R4:** We clarified in the text that the model was run until 31 December 2020.

**C5:** Line 140, Table 2: I suggest adding the temporal resolution of the data as a column of the data in this table (when applicable). Edit width for spatial resolution column.

**R5:** We have modified Table 2 (now: Table 1) to clarify the temporal resolution.

**C6:** Line 189: Same comment as line 123, add full time period of simulation.

**R6:** We clarified in the text that the model was run until 31 December 2020

**C7:** Line 236: To clarify, what was the total number of events modeled for each type of event? This modeling effort is impressive, and the number of total events modeled (prior to filtering based on impacts) would be helpful to highlight more clearly in the methods but also in the introduction or even abstract.

**R7:** We added the information on the number of events by type at the different stages to Table 3 (now: Table 2).

**C8:** Line 302, Table 4: Either in the text or as a column in the table, it would be helpful to explicitly state which classes are included in the final catalogue. If all classes are included in the catalogue that would also be helpful to state in the text. Edit width for class column.

**R8:** We clarified now in the text at all classes are included in the final catalogue, even false positives, so that users of the dataset can decide whether to use all data or limit it to certain classes.

**C9:** Line 314: How do these thresholds compare with the thresholds mentioned in Table 3?

**R9:** The thresholds were defined in previous research (HANZE v1, Paprotny et al. 2018) based on analysis of reported flood impact data to avoid inclusion of insignificant, highly localized events that would not be reproducible in pan-European flood models. The potential impact thresholds had to be much higher as the potential flood catalogue does not include flood protection, and was devised iteratively to maximize the coverage of historical events without creating too many non-impact events.

**C10:** Line 386: Text has values 11.7%, 5.4% and 3.7% for each event but it might also be helpful here to give the total number of events by event type. I would suggest including the total number of events modeled by type and then the total number of events included in the final catalogue by event type. These numbers are present throughout the text but highlighting them more explicitly (whether in this section or in introduction) would help demonstrate the scale of modeling efforts completed for this paper.

**R10:** We highlighted more clearly in the text the absolute number of events by type and class. We included additional information on the number of events in Table 3 (now: Table 2). Again, all classes are included in the published data of 15,000 events, which as highlighted in response to comment 7, was filtered from almost 26,000 events to include only those with significant potential to cause impacts.

**C11:** Line 388: The values referenced in this line are the addition of pie chart slices in Figure 2. It might be helpful to create a 4th pie chart that has the classification breakdown by all event types.

**R11:** We have added an additional graph as panel (d) to include all event to Figure 2 (now: Figure 3).

**C12:** Table 8: I would suggest changing the ratio of affected population in the 'Reported: Satellite' column to 'Satellite: Reported' to be comparable with the 'Modeled: Reported' column. Edit width of HANZE ID column in table.

**R12:** We reversed the ratios in 'Reported: Satellite' because satellite mostly showed much lower persons affected, opposite to modelled data. Therefore, it's easier to compare the ratios this way.

**Reviewer #2**

**C1:** I think it would be helpful to be more explicit about what the modelled events actually are and how they can be validated with HANZE. It is quite a forgiving test of a model framework to reward the replication of an event at such a coarse level (NUTS3), and so it would be useful to get some commentary on the physical reality in the event set as a whole and what the pairing/comparison process does and does not tell us. For example, are the August 2002 floods well replicated by flooding the wrong regions but biases of different sign cancelling out?

**R1:** In our model framework, there are no 'wrong' regions as such, as we want to include both regions were the flood impact happened and those were they did not despite the event happening there as well from only a hydrological perspective. We flag false positives from such hydrological perspective only at event level, as it would require far more detailed data collection (if they available at all) to determine if in individual regions constitute hydrological 'false positives'. Also, our comparison in 3.3.3

was limited the regions of the impacts known from (HANZE), there it shows the accuracy not affected by 'false positive' regions.

**C2:** There's no need to shy away from the inevitable subjectivity involved in creating these datasets. It could be clearer which data & method choices are well grounded, versus those which are based on assumptions or judgements. Elements such as the 2-day break between events, the various thresholds, the NUTS3 spatial limits on events: it could be clearer why these are chosen.

**R2:** Unfortunately, we did not save our test runs to show the statistics, but rerun parts of our integrated modelling chain with modified parameters to highlight how those choices affect the aggregation of events in particular. Changing the 98$^{th}$ percentile to 96$^{th}$ or 99$^{th}$ for riverine and compound events made only marginal (<2%) difference in the number regional-level events. However, when aggregated no national level, the higher threshold increased the number of events by 12%, while the lower threshold created 15% less events. Similarly, reducing the aggregation threshold to zero days when combining the regional events to national events created 11% less events, when increasing it to 3 days generated 10% more events.

**C3:** I am not sure I agree with the description of 'compound' events. To me, one hazard has to affect another to describe an event as compound. The events in this study are generated with riverine and coastal hazards independent from one another and so can only really be described as 'co-occurring'.

**R3:** We agree with the reviewer, in our view a compound events need require interaction of the riverine and coastal drivers. However, we only determine this interaction in context of impacts. In the HANZE database, the events are considered 'compound' only if such an interaction was judged to be relevant for the outcome of the flood. Otherwise, it was classified as a single-driver flood. In accordance to this, if a potential 'compound' flood is indicated in the catalogue, but impacts can be attributed, say, only to a coastal flood, the compound event was classified as 'no impact' (category C), the corresponding coastal event as impactful (category A or B) and the corresponding riverine event as 'no impact' (category C). Similarly, if a single-driver event was found to be a 'false positive' (category E), the corresponding compound event was also classified as 'false positive'. We have added this information to section 2.4.3.

**C4:** What is 'persons affected' in HANZE? Is an intersection of population and modelled flood data the same as a report of persons affected (who may be 'affected' because their road flooded and they couldn't go to work, but weren't flooded themselves). I'm not sure this quantity is particularly helpful, and cannot be modelled when only considering direct impact.

**R4:** In HANZE 'persons affected' refer to the number of people whose houses were flooded, or the number of persons evacuated if the preferred statistic is not available. We updated the definition in section 2.4.3 to make this clearer. We assume that all resident population within the potential flood zone as affected and corresponding to the HANZE statistic. This approach is likely to overestimate the affected numbers if houses are adapted to low flood depths, but still enables to differentiate the damage potential between events.

**C5:** I think the paper is long enough, but the authors could consider adding (or discussing) the sensitivity of some of their choices. Would the conclusions change entirely if you bumped the quantile threshold to 99.9%, the depth threshold to 30 cm, the window between events to 3 days? It is important for a reader to understand that only a snapshot of possible results are presented.

**R5:** As noted in R2, a higher quantile or window thresholds caused less aggregation of events, creating 10-12% more events. This leads to more cases where HANZE events are split into multiple modelled

events. As we highlight in section 2.4.2, several objectives were relevant when choosing the optimal thresholds that are specific for this study.

**C6:** The lack of flood protection data is a real problem, but I think it is conjecture to say or imply that the disparities between model and reality can be explained (solely) by flood defences. We don't actually know this, and I feel the authors use this slightly as a convenient excuse. Similarly, the idea that flood protection standards can be back-calculated on the basis of very coarse analysis of flood events is not strictly true. All such an approach would create is effective calibration parameters, which in reality will compensate for other model errors. The same can be said of the potential risk component attribution use cases. It would be helpful if the authors unstitch some of these points in their concluding remarks.

**R6:** We would like to highlight that we realize the accuracy problem and do not aim to be able to reconstruct flood protection levels below the level of NUTS3 regions. We didn't include flood defenses because including too high standards could filter out events that caused real-life impacts. We aimed at identifying the impacts based on the observations as much as possible. At the same time, the difference between observed and potential impacts is not only due to flood protection, but also the local level of vulnerability. We highlighted this issue in the conclusions, as they are indeed important for users of the data. We might add that we also explore both flood protection and vulnerability modelling based on the catalogue in detail in a follow-up preprint (https://doi.org/10.21203/rs.3.rs-4213746/v1 ), where we show that reconstructing both on our empirical data is possible, even if on a more aggregated (NUTS3) level.

**C7:** Again, the lack of flood protection data does make any attempt at validation almost meaningless (the authors acknowledge this). I think there would be scope for more detailed evaluation for select areas/regions where understanding of flood protection standards is relatively well constrained (e.g. parts of UK or Germany?) – this could give confidence that the framework as a whole is performative in spite of data limitations. Similarly, we know the Global Flood Database from Tellman et al. is very difficult to apply for event validation: it just does not contain visually realistic flood events in many cases. I think that section would be better termed a 'comparison' rather than 'validation', and the authors could look to validate against more detailed observational data in select countries (there are at least some datasets out there).

**R7:** The validation is indeed difficult due to both the framing of our study and data availability. As we look at potential impacts, useful validation would require rather the opposite: information from floods which were not so much constrained by flood defences. Also, we do realize the severe limitation of remote-sensed floods, however they are commonly used in the field, very often with the incorrect assumption that they represent the 'ground truth'. We have changed the name of section 3.3.3 to "Comparison of flood footprints" and changed mention to it in the text so that it doesn't imply validation.

**C8:** Section 1: Very minor comment, but the Rentschler et al. (2023) reference is a replica of the original study by Andreadis et al. (2022) (doi:10.1088/1748-9326/ac9197) and so it would be best to cite the latter.

**R8:** We were not aware of that study and we added the reference to it.

**C9:** Section 2.2.3: Could you describe in a bit more detail what the 'coastal segments' represent and therefore whether the derived footprints would have a realistic spatial structure?

**R9:** The segments are of variable length, depending on the complexity of the coastline. They represent no more than 25 km of the coast (if completely straight), but usually about 15 km. They stretch up to 100 km inland, but far less for more complex areas such as deltas, estuaries, fjords, islands etc. We added this information to the text.

**C10:** What is the rationale for the 99.6th percentile (compared to the 98th for riverine)? What is the rationale for the 2-day break for event separation?

**R10:** The thresholds were empirically derived with the separation threshold partly to provide about 5 potential flood events per year, as in Vousdoukas et al. (2016a), but also to maximize consistency with observed flood impact catalogue (as described in R2). As for the difference between coastal and riverine percentiles, they are due mainly to the different temporal resolution of the models (1-hourly for coastal and 6-hourly for riverine).

**C11:** Does this method mean that an event cannot occur across different NUTS3 regions?

**R11:** Not exactly, the events are first detected independently per each NUTS3 region. Only after deriving the flood zones per NUTS3-event, they are aggregated into one event if multiple NUTS3-events co-occur in time. However, this only includes NUTS3 regions within one country (at present-day boundaries), hence there are no transnational events in the catalogue.

**C12:** Section 2.3.4: I don't follow the water depth extrapolation procedure for when the return period is less than 10 years. Is it realistic to extrapolate the slope from (e.g.) the 10 and 20 year depth (if that is what is meant)?

**R12:** We indeed forgot to mention in 2.3.4 that we didn't use return periods here, but river discharge scenarios directly. But, due to the logarithmic nature of the relationship between river discharge and water level, we used the natural logarithm of discharge as basis of extrapolation. This might somewhat overstate water depths at lowest scenarios, however we found that assuming zero water depth at 2-year return period and using that for interpolation with the 10-year water depth often led to very low potential impacts and very limited impacts zones compared to the observed data in HANZE database. We have added this information to the text.

**C13:** Section 3.3.1: Why is relative discharge more important than absolute? If so, one could reasonably expect the validation to focus on the discharge which drove the flood maps rather than the event clustering.

**R13:** In determining the return periods of peak discharge, which is used to identify potentially impactful floods, the overall bias of the model is less relevant than the ability to correctly model the magnitude of flood waves relative to normal conditions. Though the riverine flood maps were not calculated with the same discharge simulation as we used, they still used an earlier version of the same pan-European model with similar meteorological forcing, therefore they should mostly have similar biases.

**Editor**

**C1-2:** Can you mention how the coastal time series was detrended (line 145)? In general, can you provide a simple explanation about why detrending is necessary (both for the river discharge and the coastal water levels).

**R1-2:** We added information to the text that the detrending was applied so that the events are distributed evenly across the 71-year time series, so they could be used in extreme value analysis, which requires stationary data.

**C3:** In Table 1 (which will be converted to a diagram as you mentioned), the text "TWL/Q" is confusing at the step "Estimating flood footprint". I believe the authors mean "TWL or Q"?

**R3:** We agree it should be "TWL or Q". We have changed this in the new workflow figure.

**C4:** In Section 2.3, am I understanding correctly that the discharge and thus the riverine model is obtained from Tilloy et al. (2024)? This was clear to me only later in the text (line 206). If the river discharges is used directly from this dataset, it would be useful to add an introduction sentence mentioning this right at the beginning of Section 2.3. If this dataset is different than the one from Tilloy et al. (2024) then again an introduction sentence explaining the main difference is welcome to help the reader.

**R4:** We have modified the text to make clear in 2.3.2 that we only repeat the methodology of Tilloy et al. (2024), not carry out new simulations, which was indeed confusing in the original text.

**C5:** Can you provide a simple explanation about the fit of the parameters for the General Pareto Distribution (i.e. according to which criteria, using which package etc.)

**R5:** Different software was used for the coastal and riverine analysis (Matlab's fitdist and Python's SciPy, respectively). Both use MLE and threshold parameters defined in the previous steps (the 99.6$^{th}$ and 98$^{th}$ percentiles, respectively). We added the information to sections 2.2.4 and 2.3.4.

**C6:** Can the authors provide a statement about the code availability?

**R6:** We have split the "Data & code availability" into two, highlighting the main code repository. However, due to many models and datasets used, details are in Appendix A3.

**Self-corrections**

The online datasets have been updated due to detected errors in the data, mostly thanks to feedback from users of the data. However, all errors were related to the preparation of the data for upload, and did not in any case affect the results of the study, or the data presented on the webpage (https://naturalhazards.eu/ ). In the dataset of list "A" events, some NUTS region codes were rendered incorrectly in the CSV file. In the dataset of list "B" events, one event start date and one country code was incorrect. Finally, in the online catalogue of flood footprints by accident excluded all coastal flood events. We have updated the links to data in Table A3.

In "Data & code availability", we added the missing reference to https://naturalhazards.eu/, where the flood catalogue can be viewed, filtered, sorted and visualized.

We also updated references to three papers which have received their final publication after the original submission of this manuscript was made.